# Brain-restricted mTOR inhibition with binary pharmacology

Ziyang Zhang[1], Qiwen Fan[2,3], Xujun Luo[2,3], Kevin Lou[1], William A. Weiss[2,3,4,5] & Kevan M. Shokat[1✉]

On-target–off-tissue drug engagement is an important source of adverse effects that constrains the therapeutic window of drug candidates[1,2]. In diseases of the central nervous system, drugs with brain-restricted pharmacology are highly desirable. Here we report a strategy to achieve inhibition of mammalian target of rapamycin (mTOR) while sparing mTOR activity elsewhere through the use of the brain-permeable mTOR inhibitor RapaLink-1 and the brain-impermeable FKBP12 ligand RapaBlock. We show that this drug combination mitigates the systemic effects of mTOR inhibitors but retains the efficacy of RapaLink-1 in glioblastoma xenografts. We further present a general method to design cell-permeable, FKBP12-dependent kinase inhibitors from known drug scaffolds. These inhibitors are sensitive to deactivation by RapaBlock, enabling the brain-restricted inhibition of their respective kinase targets.

Administration of a small-molecule drug often leads to systemic pharmacological effects that contribute to both efficacy and toxicity and define its therapeutic index. Although off-target effects may be mitigated by chemical modifications that improve the specificity of the drug, on-target–off-tissue toxicity represents a unique challenge that requires precise control over tissue partitioning. On-target–off-tissue toxicities affect commonly used drugs such as statins (myopathy caused by inhibition of HMG CoA reductase in skeletal muscle)[1] and first-generation antihistamines (drowsiness caused by blockade of the $H_1$ receptor in the brain)[2] and can sometimes preclude the safe usage of otherwise effective therapeutic agents. Chemical inhibitors of mTOR provide a case in point. Although both allosteric and orthosteric mTOR inhibitors have been investigated in numerous diseases of the central nervous system (CNS) including tuberous sclerosis complex[3–5], glioblastoma[6–9] and alcohol use disorder[10–12], systemic inhibition of mTOR is associated with various dose-limiting adverse effects: immune suppression, metabolic disorders and growth inhibition in children[13,14]. If the pharmacological effects of these mTOR inhibitors could be confined to the CNS, their therapeutic window could be substantially widened. Here we present a chemical strategy that allows brain-specific mTOR inhibition through the combination of two pharmacological agents: a brain-permeable mTOR inhibitor (RapaLink-1) whose function requires the intracellular protein FK506-binding protein 12 (FKBP12), and a brain-impermeant ligand of FKBP12 (RapaBlock). When used in a glioblastoma xenograft model, this drug combination drove tumour regression without detectable systemic toxicity. We further demonstrate that this strategy can be adapted to achieve brain-specific inhibition of other kinase targets by developing a method to convert known kinase inhibitors into FKBP12-dependent formats.

Therapeutic targeting of mTOR kinase can be achieved both allosterically and orthosterically. The first-generation mTOR inhibitors rapamycin and its analogues (rapalogues) bind to the FK506 rapamycin-binding (FRB) domain of mTOR as a complex with the intracellular protein FKBP12, resulting in substrate-dependent allosteric inhibition of mTOR complex 1 (refs. [15–17]). Second-generation mTOR kinase inhibitors (TORKi) directly bind in the ATP pocket of mTOR and inhibit the activity of both mTOR complex 1 and complex 2 (refs. [18–20]). The third-generation mTOR inhibitor RapaLink-1 is a bitopic ligand that simultaneously engages the ATP pocket and an allosteric pocket (the FRB domain) of mTOR, achieving potent and durable inhibition of its kinase activity[21]. Despite its large molecular weight (1,784 Da), RapaLink-1 is cell and brain permeant and has shown enhanced in vivo efficacy in driving glioblastoma regression compared with earlier mTOR inhibitors[7].

Because RapaLink-1 contains a TOR kinase inhibitor (TORKi) moiety (Fig. 1a, grey shading) that binds in the ATP pocket, we wondered whether the interaction of RapaLink-1 with FKBP12 is essential for its inhibition of mTOR. We performed in vitro kinase assays with purified mTOR protein and found that RapaLink-1 exhibited identical half-maximal inhibitory concentration ($IC_{50}$) values as MLN0128 (the TORKi portion of RapaLink-1) whether FKBP12 was present (Fig. 1b), indicating that RapaLink-1 can engage the active site of mTOR independent of FKBP12 in a cell-free setting. However, when we tested RapaLink-1 in cells, we observed a strong dependency on FKBP12 for mTOR inhibition. In K562 cells expressing dCas9-KRAB, CRISPRi-mediated knockdown of the gene encoding FKBP12 with two distinct single guide RNAs (sgRNAs) drastically impeded its cellular activity (Fig. 1c). This effect was even more pronounced when we used FK506 (a high-affinity natural ligand of FKBP12) to pharmacologically block the ligand-binding site of FKBP12. Similar FKBP12 dependence of RapaLink-1 was observed when we monitored mTOR signalling by western blot (Fig. 1d; see also ref. [21]). Whereas 10 nM RapaLink-1 reduced phospho-S6 (S240/244) and phospho-4EBP (T37/46) to almost undetectable levels, the combination of 10 nM RapaLink-1 and 10 μM FK506 showed no effect on either marker.

[1]Department of Cellular and Molecular Pharmacology, Howard Hughes Medical Institute, University of California, San Francisco, CA, USA. [2]Helen Diller Family Comprehensive Cancer Center, San Francisco, CA, USA. [3]Department of Neurology, University of California, San Francisco, CA, USA. [4]Department of Pediatrics, University of California, San Francisco, CA, USA. [5]Department of Neurological Surgery, University of California, San Francisco, CA, USA. ✉e-mail: kevan.shokat@ucsf.edu

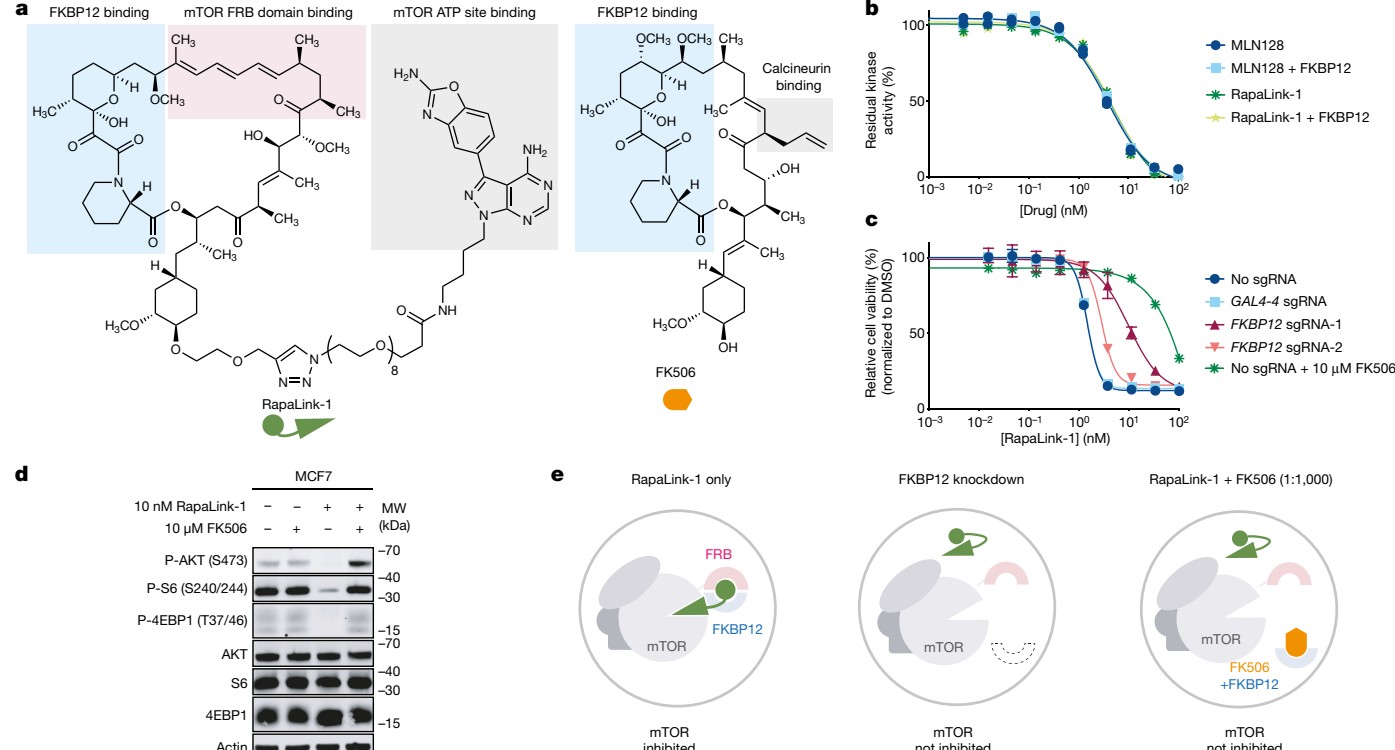

**Fig. 1 | RapaLink-1 is a potent mTOR inhibitor that requires FKBP12 for its cellular activity. a**, Chemical structures of RapaLink-1 and FK506. **b**, Inhibition of mTOR activity by MLN128, RapaLink-1 or rapamycin in the presence or absence of 10 μM FKBP12 in the in vitro kinase assay (*n* = 2; data are plotted as individual points). **c**, K562 CRISPRi cells transduced with sgRNAs targeting *GAL4-4* (control) or *FKBP12* were treated with RapaLink-1 and cell proliferation was assessed after 72 h. In the last listed condition, cells were transduced with

sgRNA but treated with RapaLink-1 in the presence of 10 μM FK506 (*n* = 3). Data are presented as mean ± s.d. **d**, Immunoblot analysis of mTOR signalling in MCF7 cells treated with DMSO, RapaLink-1, FK506 or a combination of RapaLink-1 and FK506. Results shown are representative of three independent experiments. Actin was the loading control. For gel source data, see Supplementary Fig. 1. MW, molecular weight. **e**, Schematics of the proposed working model. The grey circles indicate cellular membranes.

These results highlight that although FKBP12 is not essential for RapaLink-1 to bind to and inhibit the active site of mTOR in vitro, it is required for cellular activity, probably serving as an intracellular sink for RapaLink-1 to accumulate in the cell. We probed this possibility using a structural analogue of RapaLink-1, in which the TORKi moiety had been replaced with tetramethylrhodamine (RapaTAMRA; Extended Data Fig. 1). This fluorescent analogue of RapaLink-1 allowed us to quantify intracellular compound concentration by flow cytometry. Consistent with previous findings with RapaLink-1 (ref. [21]) and other FKBP-binding compounds[22,23], RapaTAMRA showed high cellular retention (tenfold increase in median fluorescence intensity) even after extensive wash-out, but this effect was diminished by knockdown of the gene encoding FKBP12 (Extended Data Fig. 1c). Although other factors may contribute, our data suggest that FKBP12-mediated cellular partitioning has a large role in the exceptional potency of RapaLink-1.

We next considered whether the dependence of RapaLink-1 on FKBP12 could be harnessed to achieve CNS-restricted mTOR inhibition. Specifically, selective blockade of FKBP12 in peripheral tissues with a potent, brain-impermeant small-molecule ligand would allow RapaLink to accumulate in the brain but not in peripheral tissues, resulting in brain-specific inhibition (Fig. 2a). To identify a candidate FKBP12 ligand with the suitable permeability profile (that is, cell permeable but blood–brain barrier (BBB) impermeable), we considered known natural and synthetic high-affinity FKBP12 ligands (for example, FK506 (refs. [24,25]), rapamycin[26,27] and SLF/Shield-1 (refs. [28,29])), but all of these compounds readily cross the BBB. Therefore, we synthesized a panel of derivatives of SLF and FK506 in which polar substituents (particularly hydrogen bond donors and acceptors) had been attached to the solvent-exposed parts of these two molecules,

directly opposing the empirical rules for designing BBB-permeable drugs[30,31] (Extended Data Fig. 2). In addition, modification of the C21 allyl of FK506 has the added advantage that it abolishes binding to calcineurin[32], the natural target of FK506, whose inhibition causes suppression of nuclear factor of activated T cells (NFAT) signalling and hence immunosuppression (Fig. 2b). The majority of Shield-1 and FK506 derivatives that we synthesized maintained potent FKBP12 binding (Extended Data Fig. 2), as measured by a competition fluorescence polarization assay[33]. We then screened these compounds (10 μM) in a cell-based assay, evaluating whether they can protect mTOR from inhibition by RapaLink-1 (10 nM) by monitoring the level of phospho-S6 by western blot. Although some of the SLF derivatives attenuated the potency of rapamycin (rescuing phospho-S6 levels), they failed to block RapaLink-1 (Extended Data Fig. 2a). Meanwhile, most FK506 analogues were effective, with some of them completely blocking RapaLink-1 from inhibiting mTOR (Extended Data Fig. 2b). This is probably due to the lower affinity of SLF derivatives than FKBP12, consistent with our computational modelling results (see Supplementary Note 1 for details). We subjected four compounds with distinct side-chain chemotypes to an additional in vivo screen, in which mTOR signalling was separately analysed in the brain and skeletal muscle tissues of mice treated with a combination of RapaLink-1 and the candidate compound (1 and 40 mg per kg, respectively). A pyridine *N*-oxide derivative of FK506 (06-041) protected peripheral tissues from RapaLink-1 but allowed potent inhibition of mTOR in the brain (Extended Data Fig. 3). We therefore chose this compound for further study and refer to it as 'RapaBlock' (Fig. 2c).

RapaBlock and FK506 bind to FKBP12 with comparable affinity (Fig. 2d; inhibition constant ($K_i$): 3.1 nM and 1.7 nM, respectively),

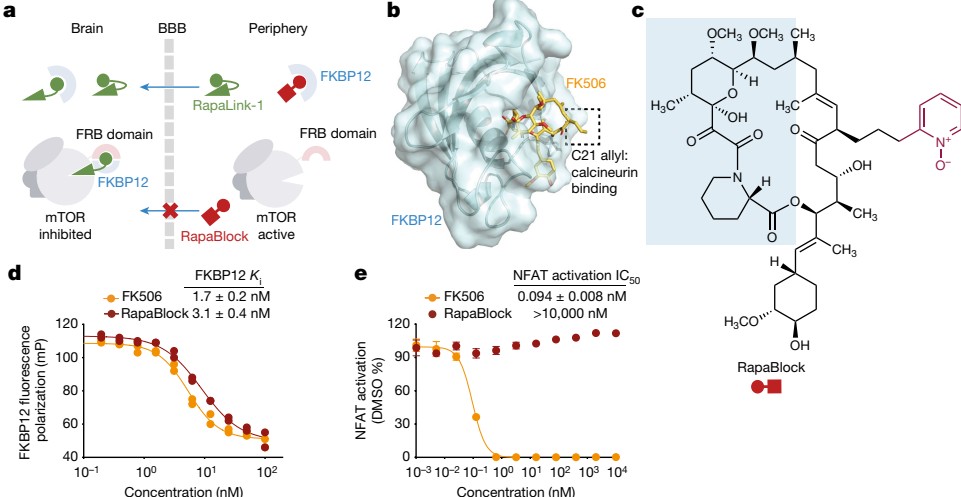

**Fig. 2 | RapaBlock is a potent, non-immunosuppressive FKBP12 ligand.**
**a**, Proposed model to achieve brain-specific mTOR inhibition through the combination of an FKBP12-dependent mTOR inhibitor (RapaLink-1) and a brain-impermeable FKBP12 ligand (RapaBlock). **b**, FK506–FKBP12 co-crystal structure (PDB: 1FKJ) showing that the C21 allyl group is solvent-exposed and its modification may lead to abolished calcineurin binding. **c**, Chemical structure of RapaBlock. The blue shaded area indicates the FKBP12-binding

moiety. The purple portion indicates the chemical modification relative to FK506. **d**, Competition fluorescence polarization assay using fluorescein-labelled rapamycin as the tracer compound ($n = 2$; data are presented as individual points). **e**, Jurkat cells expressing a luciferase under the control of the NFAT transcription response element were stimulated with phorbol myristate acetate and ionomycin in the presence of various concentrations of compounds ($n = 3$; data are presented as mean ± s.d.).

but unlike FK506, RapaBlock does not exhibit any inhibitory activity for calcineurin (Fig. 2e). In cultured cells, RapaBlock does not affect mTOR signalling by itself (up to 10 μM) but attenuates the pharmacological effects of RapaLink-1 in a dose-dependent manner (Fig. 3a). RapaBlock appears more effective at blocking rapamycin, restoring the phospho-S6 signal to the same level as untreated cells at 100:1 stoichiometry (Fig. 3d). The high affinity of RapaBlock to FKBP12 is crucial for its blocking activity—a structural analogue of RapaBlock with 100-fold reduced binding to FKBP12 failed to rescue mTOR inhibition by RapaLink-1 (Extended Data Fig. 4).

Because of the important role of mTOR signalling in driving cell growth[34], we asked whether RapaBlock can prevent RapaLink-1-mediated suppression of T cell proliferation in vitro. We stimulated human peripheral mononuclear blood cells (PBMCs) with anti-CD3 and anti-CD28 antibodies in the presence of different concentrations of RapaLink-1 and RapaBlock and assessed cell proliferation after 5 days. Whereas RapaLink-1 potently inhibited proliferation of PBMCs at nanomolar concentrations, addition of RapaBlock abolished this effect, shifting the $IC_{50}$ by more than 100-fold (Fig. 3b). We also observed higher cumulative IL-2 release in cells co-treated with RapaLink-1 and RapaBlock

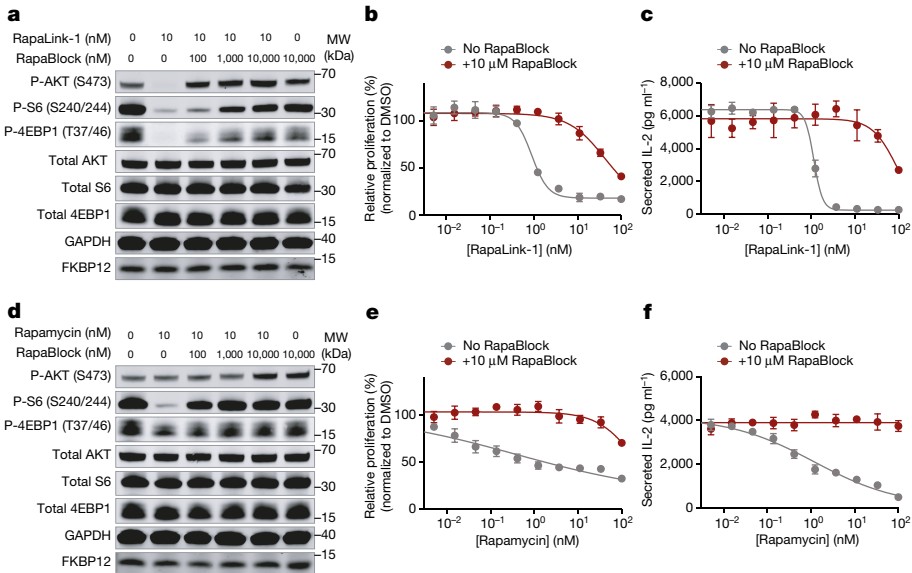

**Fig. 3 | RapaBlock protects cells from mTOR inhibition by RapaLink-1 and rapamycin. a,d**, MCF7 cells were treated with a combination of RapaLink-1 and RapaBlock (**a**), or rapamycin and RapaBlock (**d**) for 4 h, then phosphorylation of mTOR substrates were analysed by immunoblotting. Results shown are representative of three independent experiments. GDPDH was used as the loading control. For gel source data, see Supplementary Fig. 1. **b,e**, Human

PBMCs were stimulated with anti-CD3 and anti-CD28 in the presence of varying amounts of RapaLink-1 and RapaBlock (**b**) or rapamycin and RapaBlock (**e**), and cell proliferation was measured after 120 h ($n = 3$; data are presented as mean ± s.d.). **c,f**, Secretion of IL-2 in the culture supernatant of PBMCs stimulated with anti-CD3 and anti-CD28 and treated with Rapalink-1 (**c**) or rapamycin (**f**) was quantified by sandwich ELISA ($n = 3$; data are presented as mean ± s.d.).

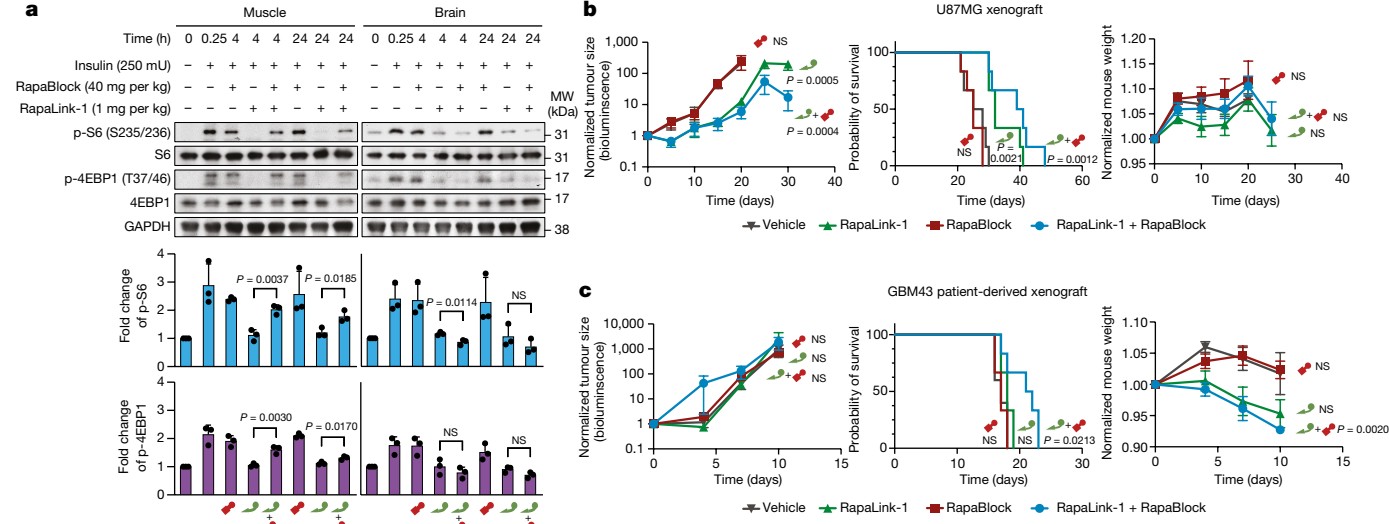

**Fig. 4 | Co-treatment with RapaLink-1 and RapaBlock allows brain-specific inhibition of mTOR. a**, Analysis of mTOR signalling in mouse whole brain and skeletal muscle after a single dose of RapaLink-1 (1 mg per kg), RapaBlock (40 mg per kg) or a combination of both. Quantified intensities are shown below each immunoblot as the fold increase relative to muscle or brain without insulin stimulation, normalized to GAPDH ($n = 3$; data are presented as mean ± s.d.). Statistical tests were performed between RapaLink-1 and RapaLink-1 + RapaBlock treatment groups at each time point (two-tailed unpaired Student's $t$-test). Immunoblot images are shown for one animal from each group. See Extended Data Fig. 5 for other replicates. All samples are derived from the same experiment. Gels and blots were processed in parallel. GAPDH was the loading control. For gel source data, see Supplementary Fig. 1. **b**, Mice ($n = 7$) bearing luciferase-expressing orthotopic glioblastoma xenografts (U87MG) were treated intraperitoneally every 5 days with: (1) vehicle; (2) RapaBlock

(40 mg per kg); (3) RapaLink-1 (1 mg per kg); and (4) RapaLink-1 (1 mg per kg) and RapaBlock (40 mg per kg). Tumour size and mouse weight were monitored every 5 days using bioluminescence imaging. Data are presented as mean ± s.d. Statistical tests were performed for each treatment group versus vehicle-treated group: two-tailed unpaired Student's $t$-test (tumour size on day 20), and log-rank test (survival). **c**, Mice ($n = 5$) bearing luciferase-expressing orthotopic glioblastoma xenografts (GBM43) were treated intraperitoneally every 5 days with: (1) vehicle; (2) RapaBlock (60 mg per kg); (3) RapaLink-1 (1.2 mg per kg); and (4) RapaLink-1 (1.2 mg per kg) and RapaBlock (60 mg per kg). Tumour size and mouse weight were monitored every 3 days using bioluminescence imaging. Data are presented as mean ± s.d. Statistical tests were performed for each treatment group versus vehicle-treated group: two-tailed unpaired Student's $t$-test (tumour size on day 10), and log-rank test (survival). NS, not significant.

than cells treated with RapaLink-1 alone (Fig. 3c). Consistent with our observation of mTOR signalling, RapaBlock appeared more effective at protecting PBMCs from rapamycin-mediated growth inhibition (Fig. 3e,f). Together, these data demonstrate that by competitively binding to intracellular FKBP12, RapaBlock renders RapaLink-1 and rapamycin incapable of inhibiting mTOR.

To examine whether the combination of RapaLink-1 and RapaBlock allows brain-specific inhibition of mTOR in vivo, we treated healthy BALB/c[nu/nu] mice with RapaLink-1 (1 mg per kg) or a combination of RapaLink-1 (1 mg per kg) and RapaBlock (40 mg per kg), stimulated mTOR activity with insulin (250 mU) after 4 h or 24 h, and analysed dissected tissues by immunoblot (Fig. 4a and Extended Data Fig. 5). RapaLink-1, when used as a single agent, potently inhibited mTOR signalling in both skeletal muscle and brain tissues, as revealed by the reduced phosphorylation of S6 and 4EBP1. However, the combination of RapaLink-1 and RapaBlock exhibited remarkable tissue-specific effects; although mTOR activity in the brain was inhibited at a comparable level to mice treated with RapaLink-1 only, mTOR activity in skeletal muscle was not inhibited. This tissue specificity has allowed us to inhibit mTOR activity in the mouse brain in chronic treatments without the common on-target side effects of mTOR inhibitors, such as body weight loss, impaired glucose metabolism and liver toxicity[35].

RapaLink-1 has been shown to be efficacious in orthotopic mouse models of glioblastoma, but inhibition of mTOR in the periphery does not contribute to efficacy. We asked whether our combination regimen, lacking the ability to inhibit mTOR activity in peripheral tissues, could retain the efficacy of RapaLink-1 in treating glioblastoma. We established orthotopic intracranial xenografts of U87MG cells expressing firefly luciferase in nude mice and treated these mice

with intraperitoneal injections of RapaLink-1 (1 mg per kg), RapaBlock (40 mg per kg) or a combination of both every 5 days. All treatments were well tolerated, and no significant changes in body weight were observed (Fig. 4b and Extended Data Fig. 6a). RapaLink-1, both as a single agent or in combination with RapaBlock, significantly suppressed tumour growth and improved survival, whereas RapaBlock alone had no significant effect on either. We next tested our treatment strategy in mice with intracranial xenografts of GBM43, a highly aggressive patient-derived glioblastoma model[36]. Although none of the conditions suppressed the rapid growth of the tumours, the combination of RapaLink-1 (1.2 mg per kg) and RapaBlock (60 mg per kg) conferred a survival benefit (Fig. 4c and Extended Data Fig. 6b).

Having established a binary therapeutic approach to achieve brain-specific inhibition of mTOR, we wondered whether the same strategy could be adapted to other drugs for which brain-restricted pharmacology would be desirable. One critical challenge is that for this approach to be generalizable, the 'active' component (for example, RapaLink-1) must be dependent on the availability of FKBP12. Drugs with this property are rare; few, if any besides rapamycin and FK506, are known. Our earlier investigation of RapaLink-1 (Fig. 1) and RapaTAMRA (Extended Data Fig. 1) led us to hypothesize that other bifunctional molecules consisting of a FKBP12-binding moiety and a kinase inhibitor moiety could be conditionally active and amenable to modulation with RapaBlock.

We first tested our hypothesis with GNE7915, a potent and specific inhibitor of leucine-rich repeat kinase 2 (LRRK2) under pre-clinical investigation for the treatment of Parkinson disease[37]. As gain-of-function mutations of LRRK2 are strongly associated with hereditary and sporadic forms of Parkinson disease, LRRK2 kinase

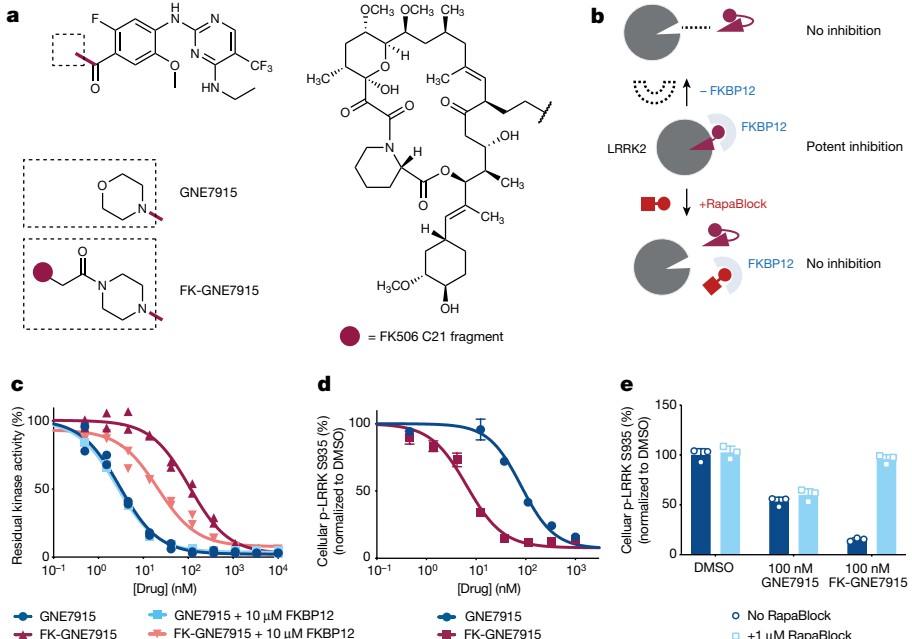

**Fig. 5 | Programmable kinase inhibition with FKBP-dependent kinase inhibitors and RapaBlock. a**, Structures of GNE7915 and FK-GNE7915. **b**, Proposed working model for FKBP-dependent kinase inhibitors. **c**, Kinase inhibition in the absence or presence of supplemented 10 μM recombinant FKBP12 protein (*n* = 2; data are shown as individual points). **d,e**, RAW264.7 cells were treated with GNE7915, FK-GNE7915 and/or RapaBlock, and phospho-LRRK2 (S935) was analysed by time-resolved FRET using epitopically orthogonal antibodies for LRRK2 and p-LRRK2 (S935) (*n* = 3; data are presented as mean ± s.d.). See Supplementary Information for experimental details.

inhibition has been pursued as a potential therapeutic strategy[38,39]. However, recent studies demonstrating that systemic LRRK2 inhibition with small-molecule inhibitors induced cytoplasmic vacuolation of type II pneumocytes have suggested a potential safety liability for these compounds[40,41]. We synthesized FK-GNE7915 by chemically linking the pharmacophores of FK506 and GNE7915 with a piperazine group (Fig. 5a). In in vitro LRRK2 kinase assays, FK-GNE7915 was an inferior inhibitor to GNE7915 (Fig. 5c; IC$_{50}$ of 107 nM and 3.0 nM, respectively), although its activity was potentiated by including 10 μM FKBP12 in the assay (IC$_{50}$ of 21 nM). However, in a cellular assay in which we quantified phospho-LRRK2 (S935) levels as a marker for LRRK2 inhibition, FK-GNE7915 was more potent than the parent compound GNE7915 by more than tenfold (Fig. 5d; IC$_{50}$ of 6.7 nM and 81 nM, respectively). The juxtaposition of these two results suggested a role of the FK506 moiety in the enhanced cellular potency of FK-GNE7915. We reasoned that the high-affinity FK506–FKBP12 interaction can promote the intracellular accumulation of FK-GNE7915 similar to the case of RapaLink-1 and confirmed this using a bifunctional fluorescent probe: FK-TAMRA (Extended Data Fig. 7). This feature allowed us to programme LRRK2 inhibition by controlling the availability of FKBP12; whereas treatment with 100 nM FK-GNE7915 reduced the level of phospho-LRRK2 (S935) to 15% of the basal level, the combination of 100 nM FK-GNE7915 and 1 μM RapaBlock had no effect on LRRK2 activity (Fig. 5e).

The successful conversion of GNE7915 into a FKBP-dependent LRRK2 inhibitor by simply linking it to an FK506 fragment prompted us to evaluate the generality of this strategy. We explored several kinase inhibitors that are being investigated for CNS diseases: dasatinib (Src family kinase inhibitor for glioblastoma)[42,43], lapatinib (EGFR/HER2 inhibitor for glioblastoma)[44,45] and prostetin (MAP4K4 inhibitor for amyotrophic lateral sclerosis and Alzheimer disease)[46]. In all three cases, chemically linking the kinase inhibitor to FK506 yielded bifunctional molecules that retained the kinase inhibitory activities of the parent molecules (Extended Data Figs. 8–10). For FK-dasatinib, we also examined its target specificity using both biochemical (Invitrogen SelectScreen Kinase Profiling; Supplementary Table 1) and live-cell

kinase profiling[47]. Among the reported targets of dasatinib, Src family kinases were potently inhibited by both dasatinib and FK-dasatinib, whereas a few tyrosine kinases (for example, DDR2) exhibited differential susceptibility to these two inhibitors (Extended Data Fig. 8c). Despite their large molecular weights, these bifunctional molecules were active in cells with comparable potencies to their parent compounds and sensitive to deactivation by RapaBlock. Although these examples represent a limited set of kinase inhibitors with potential CNS disease indications, it is conceivable that more drugs can be similarly configured with programmable pharmacology without losing cellular potency.

## Conclusion

By exploiting the unique functional dependence of rapamycin analogues on FKBP12, we have developed an approach to achieve brain-specific mTOR inhibition through the simultaneous administration of a potent mTOR inhibitor (RapaLink-1) and a cell-permeable, BBB-impermeable ligand of FKBP12 (RapaBlock). Tissue-restricted mTOR inhibition enabled by this binary pharmacology strategy reduced the toxicity in peripheral tissues but maintained therapeutic benefits of RapaLink-1 in glioblastoma xenograft models. On the basis of these findings, it seems reasonable to anticipate that the same drug combination may be of broader value in other CNS diseases driven by dysregulated mTOR activity such as alcohol use disorder[35].

The applicability of our approach extends beyond mTOR inhibition. We show that chemically linking ATP-site kinase inhibitors to FK506 through solvent-exposed groups leads to a new class of cell-permeable kinase inhibitors whose activity depends on the abundant endogenous protein FKBP12. These inhibitors are characterized by their ability to mediate the formation of a ternary complex of the drug, the target kinase and FKBP12, as well as their amenability to activity modulation by RapaBlock. Further in vivo studies and medicinal chemistry are necessary to assess and optimize these compounds on a case-by-case basis, but our initial investigations show that it is feasible to attain

brain-selective kinase inhibition of LRRK2 using a FKBP-dependent kinase inhibitor (FK-GNE7915) and RapaBlock.

We have investigated the mechanism by which RapaBlock controls the cellular activity of RapaLink-1 as well as other FKBP12-dependent kinase inhibitors. Our current data have revealed at least two roles of cellular FKBP12 in the function of RapaLink-1 (and other FKBP12-binding compounds). First, we have shown that FKBP12 serves as a reservoir to retain and accumulate RapaLink-1 inside the cell, achieving exceptional cellular concentration. Second, for most FKBP12-dependent kinase inhibitors that we have investigated, FKBP12 improves their potency in cell-free assays. Although we have not obtained direct evidence, we hypothesize that in aqueous solutions, a bifunctional compound built with flexible linkers between hydrophobic pharmacophores will mainly adopt a binding-incompetent conformation to minimize hydration penalty ('hydrophobic collapse'), and binding of FKBP12 will expose the inhibitor moiety to enable target inhibition. RapaBlock impedes both of these processes by occupying the ligand-binding site of FKBP12. Although further investigation is clearly necessary to elucidate how these high-molecular-weight compounds enter cells and whether FKBP12 participates in the binding interaction with the target protein, we believe that this system provides a generalizable approach for programmable kinase inhibition.

Combining two pharmaceutical agents to achieve tissue-selective therapeutic effects has been previously used in drugs approved for clinical use or in development. Examples include levodopa–carbidopa (BBB-permeable dopamine precursor–BBB-impermeable DOPA decarboxylase inhibitor) for Parkinson disease[48], conjugated oestrogens–bazedoxifene (BBB-permeable oestrogen–BBB-impermeable oestrogen receptor modulator) for post-menopausal hot flashes and osteoporosis[49], and donepezil–solifenacin (BBB-permeable acetylcholinesterase inhibitor–BBB-impermeable anticholinergic) for Alzheimer disease[50]. Our approach differs from these precedents in that it does not involve two drugs with counteracting effects on the same target or pathway; instead, RapaBlock controls tissue specificity by directly attenuating the activity of the kinase inhibitor. The present system therefore has the advantage of being adaptable for a multitude of targets, only requiring an invariant RapaBlock molecule and a FKBP12-dependent inhibitor that can be readily designed based on the structures of FK506 and lead compounds. Although we have focused on protein kinases in this study, it seems reasonable to expect that the approach is also applicable to other classes of therapeutic targets, such as GTPases and histone modification enzymes.

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

## Methods

### Expression and purification of recombinant FKBP12

DNA sequences encoding full-length human FKBP12 were synthesized by Twist Biosciences and cloned into the pET47b vector using standard molecular biology techniques. Protein expression was performed in BL21(DE3) *Escherichia coli* strain. In brief, chemically competent BL21(DE3) cells were transformed with pET47b-FKBP12 and grown on LB medium plates containing 50 µg ml$^{-1}$ kanamycin at 37 °C. A single colony was used to inoculate a culture at 37 °C at 220 r.p.m. in terrific broth containing 50 µg ml$^{-1}$ kanamycin. When the optical density reached 0.6, protein expression was induced by the addition of IPTG to 1 mM. After 2 h at 37 °C, the cells were pelleted by centrifugation (6,500$g$ for 10 min) and lysed in lysis buffer (20 mM Tris 8.0, 500 mM NaCl and 5 mM imidazole) with a high-pressure homogenizer (Microfluidics). The lysate was clarified by high-speed centrifugation (19,000$g$ for 15 min) and the supernatant was used in subsequent purification by immobilized metal affinity chromatography. His-tagged FKBP12 was captured by incubation with Co-TALON resin (Clonetech, Takara Bio USA, 4 ml slurry per litre culture) at 4 °C for 1 h with constant end-to-end mixing. The loaded beads were then washed with lysis buffer (50 ml l$^{-1}$ culture) and the protein was eluted with elution buffer (20 mM Tris 8.0, 500 mM NaCl and 300 mM imidazole). The His-tag was cleaved with His-tagged HRV 3C Protease (Clonetech, Takara Bio USA, 5 U l$^{-1}$ culture) at 4 °C until liquid chromatography–mass spectrometry analysis of the reaction mixture indicated more than 95% cleavage. The reaction mixture was concentrated using a 10K MWCO centrifugal concentrator (Amicon-15, Millipore) to 20 mg ml$^{-1}$ and purified by size-exclusion chromatography on a Superdex 75 10/300 GL column (GE Healthcare Life Sciences) with SEC buffer (20 mM HEPES pH 7.5 and 150 mM NaCl). Fractions containing pure FKBP12 protein were pooled and concentrated to 20 mg ml$^{-1}$ and stored at −78 °C. This protocol gives a typical yield of 10–20 mg l$^{-1}$ culture for FKBP12.

### Determination of compound binding affinity to FKBP12

Compound binding affinity was determined using a competition fluorescence polarization assay. A fluorescent tracer molecule based on rapamycin (FITC-Rapa)[33] was synthesized in house. The assay buffer was 20 mM HEPES pH 7.5 and 0.01% Triton X-100. The dissociation constant ($K_d$) of the tracer molecule for FKBP12 was first determined by measuring fluorescence polarization (excitation of 485 nm and emission of 535 nm) at various protein concentrations and fitting the curve to a quadratic binding model. To measure compound binding affinity, mixtures with the following composition were prepared in duplicate in 96-well black opaque plates (Corning 3915): 0.5 nM FITC-Rapa, 1 nM FKBP12, 5% DMSO and 5 µM to 0.08 nM of test compound, 200 µl total volume. Fluorescence polarization was measured on a TECAN Spark 20M plate reader (excitation of 485 nm and emission of 535 nm). Data were fitted to a three-parameter sigmoidal curve to derive IC$_{50}$ values. $K_i$ of the compounds were calculated using a tool provided by S. Wang's laboratory (http://www.umich.edu/~shaomengwanglab/software/calc_ki/calc_ki.html). Data were plotted in GraphPad Prism 9.0.

### Cell culture

MCF7 cells were obtained from the American Type Culture Collection (ATCC; HTB-22) and maintained in 1:1 DMEM:F12 (Gibco) plus 10% heat-inactivated FBS (Axenia Biologix) supplemented with 4 mM L-glutamine, 100 U ml$^{-1}$ penicillin and 100 U ml$^{-1}$ streptomycin (Gibco). SK-BR-3 cells were obtained from the ATCC (HTB-30) and maintained in McCoy's 5A (Gibco) plus 10% heat-inactivated FBS supplemented with 2 mM L-glutamine, 100 U ml$^{-1}$ penicillin and 100 U ml$^{-1}$ streptomycin (Gibco). K562 CRISPRi cells were a gift from L. Gilbert and maintained in RPMI 1640 (Gibco) plus 10% heat-inactivated FBS supplemented with 2 mM L-glutamine, 100 U ml$^{-1}$ penicillin, 100 U ml$^{-1}$ streptomycin (Gibco) and 0.1% Pluronic F-68 (Gibco). RAW264.7 cells were obtained from the ATCC (TIB-71) and maintained in DMEM (Gibco) plus 10% heat-inactivated FBS supplemented with 2 mM L-glutamine, 100 U ml$^{-1}$ penicillin and 100 U ml$^{-1}$ streptomycin (Gibco). Jurkat cells were obtained from the ATCC (TIB-152) and maintained in RPMI 1640 (Gibco) plus 10% heat-inactivated FBS supplemented with 2 mM L-glutamine, 100 U ml$^{-1}$ penicillin and 100 U ml$^{-1}$ streptomycin (Gibco). Jurkat-Lucia NFAT cells were obtained from InvivoGen and maintained in IMDM (Gibco) plus 10% heat-inactivated FBS supplemented with 2 mM L-glutamine, 100 U ml$^{-1}$ penicillin, 100 U ml$^{-1}$ streptomycin (Gibco) and 100 µg ml$^{-1}$ Zeocin (InvivoGen). HEK293T cells were obtained from UCSF Cell Culture Facility and maintained in 1:1 DMEM:F12 (Gibco) plus 10% heat-inactivated FBS supplemented with 4 mM L-glutamine, 100 U ml$^{-1}$ penicillin and 100 U ml$^{-1}$ streptomycin (Gibco). All cell lines tested mycoplasma negative using the MycoAlert Mycoplasma Detection Kit (Lonza). Cell lines from the ATCC were short random repeat (STR) profiled by the manufacturer. Jurkat-Lucia NFAT cells are a commercial cell line developed by InvivoGen and was authenticated by the manufacturer. Cell lines from UCSF Cell Culture Facility were STR profiled by the UCSF Cell Culture Facility. No further authentications were performed after the purchase. When indicated, cells were treated with drugs at 60–80% confluency at a final DMSO concentration of 1%. At the end of the treatment period, cells were placed on ice and washed once with PBS. Unless otherwise indicated, the cells were scraped with a spatula, pelleted by centrifugation (500$g$ for 5 min) and lysed in RIPA buffer (25 mM Tris pH 7.4, 150 mM NaCl, 0.1% SDS 1% NP-40 and 0.5% sodium deoxycholate) supplemented with protease and phosphatase inhibitors (cOmplete and phosSTOP, Roche) on ice for 10 min. Lysates were clarified by high-speed centrifugation (19,000$g$ for 10 min). Concentrations of lysates were determined with a protein BCA assay (Thermo Fisher) and adjusted to 2 mg ml$^{-1}$ with additional RIPA buffer. Samples were mixed with 5× SDS loading dye and heated at 95 °C for 5 min.

### Gel electrophoresis and western blot

Unless otherwise noted, SDS–PAGE was run with Novex 4–12% Bis-Tris gel (Invitrogen) in MES running buffer (Invitrogen) at 200 V for 40 min following the manufacturer's instructions. Protein bands were transferred onto 0.45-µm nitrocellulose membranes (Bio-Rad) using a wet-tank transfer apparatus (Bio-Rad Criterion Blotter) in 1× TOWBIN buffer with 10% methanol at 75 V for 45 min. Membranes were blocked in 5% BSA–TBST for 1 h at 23 °C. Primary antibody binding was performed with the indicated antibodies diluted in 5% BSA–TBST at 4 °C for at least 16 h. After washing the membrane three times with TBST (5 min for each wash), secondary antibodies (goat anti-rabbit IgG-IRDye 800 and goat anti-mouse IgG-IRDye 680; Li-COR) were added as solutions in 5% skim milk–TBST at the dilutions recommended by the manufacturer. Secondary antibody binding was allowed to proceed for 1 h at 23 °C. The membrane was washed three times with TBST (5 min for each wash) and imaged on a Li-COR Odyssey fluorescence imager. See Supplementary Table 2 for a full list of antibodies used in this study.

### NFAT activation assay

Jurkat-Lucia NFAT cells were resuspended to 2 × 10$^6$ cells per ml in fresh growth medium and dispensed in 96-well tissue culture plates (Corning 3904; 180 µl per well). DMSO solutions of test compounds at 100× the test concentrations were added and the cells were incubated for 4 h at 37 °C. A 10× stimulation solution containing phorbol myristate acetate (100 ng ml$^{-1}$) and ionomycin (10 µg ml$^{-1}$) were added to each well (20 µl per well) except for the negative control wells, which were supplemented with 20 µl medium. Cells were incubated at 37 °C for 12 h. Cells (20 µl per well) were transferred into a white opaque 96-well plate (Corning 3912). TECAN Spark 20M plate reader equipped with an auto-injection system was primed with Lucia luciferase substrate solution and set with the following parameters: 50 µl of injection volume, end-point measurement with a 4-s delay time and 0.1-s integration time. Luciferase activity was measured with the settings above and normalized to DMSO-treated, stimulated cells. Data were plotted in GraphPad Prism 9.0.

## Jurkat cell stimulation

Jurkat cells were cultured and treated as described in the 'Cell culture' section. At the end of treatment period, 1.5 ml aliquot of cells were pelleted (500g for 5 min) and washed once with serum-free RPMI (1 ml). The supernatant was removed, and the cells were resuspended in 100 µl 5 µg ml$^{-1}$ OKT3 (Invitrogen; functional grade) in RPMI and immediately placed in a 37 °C water bath. At 5 min, 25 µl 5× SDS loading buffer was added and mixed quickly with the cells. The sample was sonicated at 30% power output for 60 s (1 s on, 1 s off) using Qsonica Q500 Sonicator to shear the DNA. The samples were heated at 95 °C for 5 min and used for SDS–PAGE.

## Phospho-LRRK2 quantification by TR-FRET

Phospho-LRRK2 (S935) was quantified in drug-treated cells using a Phospho-LRRK2 (Ser935) cellular kit (Cisbio) following the manufacturer's instructions. RAW264.7 ($2 × 10^5$ cells per ml) cells were plated in six-well tissue culture plates (2 ml per well) 24 h before treatment. Cells were treated with compounds at the indicated concentrations for 2 h. Medium was removed by aspiration and the cells were rinsed with ice-cold PBS (1 ml). Cells were lysed with 100 µl Cisbio 1X lysis buffer at 23 °C directly in plate for 30 min. The lysates were transferred into microcentrifuge tubes and clarified by centrifugation (19,000g for 10 min). Clarified lysate (16 µl) was dispensed into one well of a low-volume, round-bottom 384-well plate, and 4 µl LRRK2–phospho-LRRK2 (S935) antibody master mix (Cisbio) was added to each well. The plate was incubated at 23 °C for 4 h. Time-resolved fluorescence was read on a TECAN Spark 20M plate reader with the following parameters: lag time of 60 µs; integration time of 500 µs; read A included excitation filter of 320 (25) nm, emission filter of 610 (25) nm and gain of 130; read B included excitation filter of 320 (25) nm, emission filter of 665 (8) nm and gain of 165.

The TR-FRET signal was calculated as the ratio fluorescence intensity [read B]/[read A]. Data were plotted in GraphPad Prism 9.0.

## DNA transfections and lentivirus production

HEK293T cells were transfected with standard packaging vectors using the TransIT-LT1 Transfection Reagent (Mirus Bio). Viral supernatant was collected 2–3 days after transfection, filtered through 0.45-µm polyvinylidene difluoride filters (Millipore Sigma) and frozen at −80 °C before transduction.

## Generation of sgRNA-expressing CRISPRi cell lines

sgRNA protospacers targeting *GAL4-4* (negative control sg GAACGAC TAGTTAGGCGTGTA) and *FKBP12* (FKBP12 sg1 GACGGCTCTGCCTAG TACCT and FKBP12 sg2 GCCCAGGAGACGGTGAGTAG)[51] were cloned into pCRISPRia-v2 (marked with a puromycin resistance cassette and BFP; Addgene #84832). In brief, complementary synthetic oligonucleotides (Integrated DNA Technologies) with flanking BstXI and BlpI restriction sites were annealed and ligated with BstXI- and BlpI-digested pCRISPRia-v2. The sgRNA expression vectors were packaged into lentivirus as described above. Knockdown cells were generated by transducing K562 CRISPRi (sgRNA$^-$) cells with the sgRNA expression vectors at a multiplicity of infection of less than 1 (20–40% transduction rate) with 8 µg ml$^{-1}$ polybrene. Beginning the second day following lentiviral addition, transduced (sgRNA$^+$) cells were selected using 2 µg ml$^{-1}$ puromycin (Gibco) until each cell population stably reached 95% or more BFP$^+$ (405-nm excitation laser, and 440/50-nm emission filter) by flow cytometry on an Attune NxT (Thermo Fisher Scientific).

## Cell viability assay

Cells were seeded into 96-well white flat bottom plates (1,000 cells per well) (Corning) and incubated overnight. Cells were treated with the indicated compounds in a nine-point threefold dilution series (100 µl final volume) and incubated for 72 h. In some conditions, 10 µM FK506

was uniformly added to the dilution series. Cell viability was assessed using a commercial CellTiter-Glo (CTG) luminescence-based assay (Promega). In brief, the 96-well plates were equilibrated to room temperature before the addition of diluted CTG reagent (100 µl) (1:4 CTG reagent:PBS). Plates were placed on an orbital shaker for 30 min before recording luminescence using a Spark 20M (TECAN) plate reader. Data were plotted in GraphPad Prism 9.0.

## Cellular dye (TAMRA) retention assay

Untransduced (no virus or sgRNA$^-$) and sgRNA expression vector-transduced (sgRNA$^+$) K562 CRISPRi cells were mixed in a 1:1 ratio, plated in 48-well plates and incubated overnight. Cell mixtures were treated with the indicated TAMRA-linked compounds (300 µl final volume) and incubated for 24 h. Plates were placed over ice and 200 µl of each compound–cell mixture was transferred to a 96-well U-bottom plate. Cells were pelleted at 500g for 5 min and subsequently washed 2× with ice-cold FACS buffer (PBS + 1% BSA + 0.1% NaN$_3$). Cells were resuspended in 200 µl of ice-cold FACS buffer and assessed using an Attune NxT (Thermo Fisher Scientific). Relative uptake of TAMRA-linked compounds was determined by comparing TAMRA fluorescence (561-nm excitation laser and 585/16-nm emission filter) between sgRNA$^-$ and sgRNA$^+$ cells within each well.

## In vitro kinase inhibition assay

In vitro kinase inhibition assays were performed by SelectScreen Services (Thermo Fisher Scientific) using either Z′-LYTE (mTOR, Src, Csk, EGFR, HER2 and HGK), Adapta (LRRK2) or LanthaScreen (DDR2) assay format (standardized protocols can be accessed on the SelectScreen website: https://www.thermofisher.com/us/en/home/products-and-services/services/custom-services/screening-and-profiling-services/selectscreen-profiling-service/selectscreen-kinase-profiling-service.html. When indicated, recombinant FKBP12 protein was added to the assay mixture to a final concentration of 10 µM.

## Animal studies

All animal experiments were conducted using protocols approved by the Institutional Animal Care and Use Committee (IACUC) of the University of California, San Francisco. Female athymic BALB/C$^{nu/nu}$ athymic nude mice (4–6 weeks old) were used for in vivo experiments. All mice at the Helen Diller Cancer Center UCSF were housed in individually ventilated microisolator cages. The housing racks had automatic water that was sterilized through filtration, reverse osmosis and UV light exposure. All mice were fed a standard diet that was pre-irradiated. The light–dark cycle was 12 h light–12 h dark. Temperature was maintained in the range of 68–74 °F. Humidity was maintained at 30–70%. Western blot analysis of mTOR signalling used 4–6-week-old female athymic BALB/C$^{nu/nu}$ mice (three per group; randomized) that were treated with intraperitoneal injections of vehicle (20% DMSO, 40% PEG-300 and 40% PBS (v/v)), RapaLink-1 (1 mg per kg), RapaBlock (40 mg per kg), and a combination of RapaLink-1 (1 mg per kg) and RapaBlock (40 mg per kg) for 15 min, followed by intraperitoneal injection of 250 mU insulin or saline, then euthanized 15 min later. Skeletal muscle and brain from each mouse were lysed and analysed by western blotting. For orthotopic injections and treatment studies, female BALB/C$^{nu/nu}$ mice (4–6 weeks old) were anaesthetized using isoflurane. U87MG cells ($3 × 10^5$) expressing firefly luciferase were injected intracranially (Hamilton syringe) at coordinates 2 mm anterior and 1.5 mm lateral of the right hemisphere relative to Bregma, at a depth of 3 mm. Whole-brain bioluminescence was measured for each mouse every 5 days. When bioluminescence reached $10^6$ photons per second, mice were randomized to four groups of equal mean bioluminescent signal (seven mice per group) and therapy was initiated. Groups were treated with intraperitoneal injection of vehicle (20% DMSO, 40% PEG-300 and 40% PBS (v/v)), RapaLink-1 (1.0 mg per kg), RapaBlock (40 mg per kg),

or a combination of RapaLink-1 (1.0 mg per kg) and RapaBlock (40 mg per kg) every 5 days for 25–30 days. Mice were monitored daily and euthanized when they exhibited neurological deficits or 15% reduction from initial body weight. For GBM43 xenograft studies, GBM43 cells ($1 \times 10^5$) expressing firefly luciferase were injected intracranially (Hamilton syringe) at coordinates 2 mm anterior and 1.5 mm lateral of the right hemisphere relative to Bregma, at a depth of 3 mm. Whole-brain bioluminescence was measured for each mouse every 5 days. When bioluminescence reached $10^6$ photons per seconds, mice were randomized to four groups of equal mean bioluminescent signal (five mice per group), and therapy was initiated. Groups were treated with intraperitoneal injection of vehicle (20% DMSO, 40% PEG-300 and 40% PBS (v/v)), RapaLink-1 (1.2 mg per kg), RapaBlock (60 mg per kg), or a combination of RapaLink-1 (1.2 mg per kg) and RapaBlock (60 mg per kg) every 5 days for 15–20 days. Mice were monitored daily and euthanized when they exhibited neurological deficits or 15% reduction from initial body weight. Researchers were blinded from mice group assignment for imaging luciferase activity and measuring tumour burden.

## Chemical synthesis

See Supplementary Notes 2 and 3 for details on chemical synthesis and $^1$H and $^{13}$C NMR spectra of RapaBlock.

## Reporting summary

Further information on research design is available in the Nature Research Reporting Summary linked to this article.

## Data availability

Unprocessed gel images for all immunoblots (Figs. 1d, 3a,d and 4a and Extended Data Figs. 3, 4c, 5, 8e,f and 9d) are provided in Supplementary Fig. 1. Raw data for the kinase inhibition assays, cell growth assays and xenograft studies are available as source data files. All data that support the findings of this study are available from the corresponding author on reasonable request. Source data are provided with this paper.

51. Horlbeck, M. A. et al. Compact and highly active next-generation libraries for CRISPR-mediated gene repression and activation. *eLife* **5**, e19760 (2016).

**Acknowledgements** We thank J. Koach for help with data processing; D. Wassarman for discussions; our UCSF colleague M. Evans and his group for many discussions and mouse brain imaging studies; and L. Gilbert for the generous gift of K562 CRISPRi (dCas9-KRAB) cells. Z.Z. is a Damon Runyon Fellow supported by the Damon Runyon Cancer Research Foundation (DRG-2281-17). K.L. acknowledges NIH F30CA239476. K.M.S., Q.F. and W.A.W. acknowledge NIH 1R01CA221969. K.M.S. and W.A.W. acknowledge the Samuel Waxman Cancer Research Foundation. W.A.W. acknowledges NIH U01CA217864 and Cancer Research UK A28592. K.M.S. acknowledges the Michael J. Fox Foundation P0536220, The Mark Foundation for Cancer Research and the Howard Hughes Medical Institute.

**Author contributions** Z.Z., Q.F., W.A.W. and K.M.S. conceived the project, designed and analysed the experiments, and wrote the manuscript. W.A.W. supervised the in vivo experiments. Z.Z. performed chemical synthesis, the biochemical assays and cell culture experiments, and analysed the data. K.L. performed the flow cytometry experiments and growth inhibition assays in K562 CRISPRi cell lines and analysed the data. Q.F. and X.L. performed the in vivo experiments and immunoblot analysis of the tissue samples, and analysed the data.

**Competing interests** Z.Z., Q.F., W.A.W. and K.M.S. are co-inventors on patent applications covering RapaBlock owned by UCSF. K.M.S. is an inventor on patents covering RapaLink-1 owned by UCSF and licensed to Revolution Medicines. K.M.S. receives monetary and stock compensation and is an SAB member of Genentech/Roche, Denali Therapeutics and Revolution Medicines. Kin of K.L. hold stock in and are employed by Pharmaron.

**Additional information**
**Correspondence and requests for materials** should be addressed to Kevan M. Shokat.

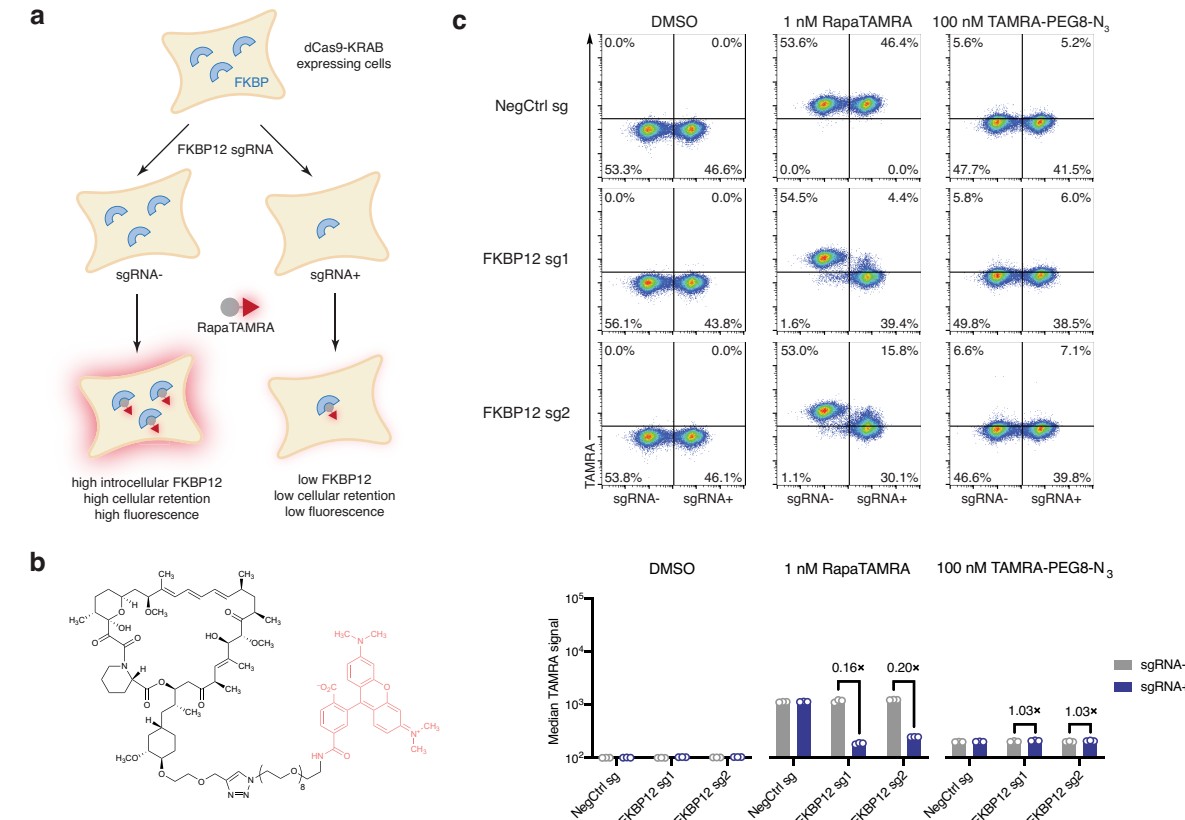

**Extended Data Fig. 1 | FKBP12-Rapamycin interaction contributes to cellular accumulation of a fluorescent analog of RapaLink-1. a**, Illustration of the flow cytometry-based assay to assess cellular accumulation of TAMRA compounds. **b**, Structure of a fluorescent probe RapaTAMRA. **c**, FKBP12 knockdown decreases cellular retention of RapaTAMRA, but not

TAMRA-PEG8-N3. Flow cytometry scatter plots shown are representative of three independent replicates, which are summarized in the bar graphs. Data are presented as mean values +/− SD with individual data points. Fold change is calculated as the ratio between means (sgRNA+/sgRNA−). See Supplementary Fig. 2 for gating strategy.

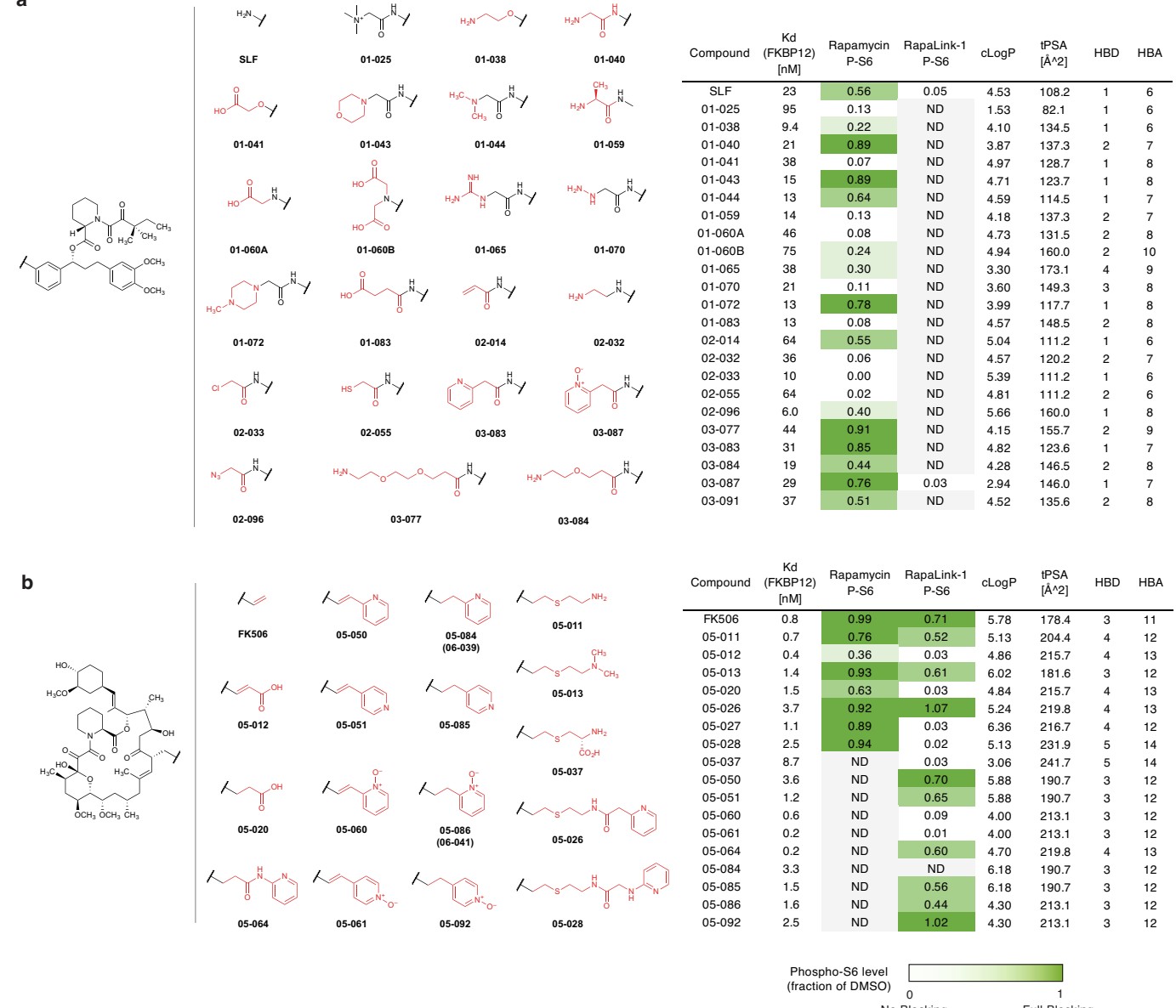

**a**

| Compound | Kd (FKBP12) [nM] | Rapamycin P-S6 | RapaLink-1 P-S6 | cLogP | tPSA [Å^2] | HBD | HBA |
|---|---|---|---|---|---|---|---|
| SLF | 23 | 0.56 | 0.05 | 4.53 | 108.2 | 1 | 6 |
| 01-025 | 95 | 0.13 | ND | 1.53 | 82.1 | 1 | 6 |
| 01-038 | 9.4 | 0.22 | ND | 4.10 | 134.5 | 1 | 6 |
| 01-040 | 21 | 0.89 | ND | 3.87 | 137.3 | 2 | 7 |
| 01-041 | 38 | 0.07 | ND | 4.97 | 128.7 | 1 | 8 |
| 01-043 | 15 | 0.89 | ND | 4.71 | 123.7 | 1 | 8 |
| 01-044 | 13 | 0.64 | ND | 4.59 | 114.5 | 1 | 7 |
| 01-059 | 14 | 0.13 | ND | 4.18 | 137.3 | 2 | 7 |
| 01-060A | 46 | 0.08 | ND | 4.73 | 131.5 | 2 | 8 |
| 01-060B | 75 | 0.24 | ND | 4.94 | 160.0 | 2 | 10 |
| 01-065 | 38 | 0.30 | ND | 3.30 | 173.1 | 4 | 9 |
| 01-070 | 21 | 0.11 | ND | 3.60 | 149.3 | 3 | 8 |
| 01-072 | 13 | 0.78 | ND | 3.99 | 117.7 | 1 | 8 |
| 01-083 | 13 | 0.08 | ND | 4.57 | 148.5 | 2 | 8 |
| 02-014 | 64 | 0.55 | ND | 5.04 | 111.2 | 1 | 6 |
| 02-032 | 36 | 0.06 | ND | 4.57 | 120.2 | 2 | 7 |
| 02-033 | 10 | 0.00 | ND | 5.39 | 111.2 | 1 | 6 |
| 02-055 | 64 | 0.02 | ND | 4.81 | 111.2 | 2 | 6 |
| 02-096 | 6.0 | 0.40 | ND | 5.66 | 160.0 | 1 | 8 |
| 03-077 | 44 | 0.91 | ND | 4.15 | 155.7 | 2 | 9 |
| 03-083 | 31 | 0.85 | ND | 4.82 | 123.6 | 1 | 7 |
| 03-084 | 19 | 0.44 | ND | 4.28 | 146.5 | 2 | 8 |
| 03-087 | 29 | 0.76 | 0.03 | 2.94 | 146.0 | 1 | 7 |
| 03-091 | 37 | 0.51 | ND | 4.52 | 135.6 | 2 | 8 |

**b**

| Compound | Kd (FKBP12) [nM] | Rapamycin P-S6 | RapaLink-1 P-S6 | cLogP | tPSA [Å^2] | HBD | HBA |
|---|---|---|---|---|---|---|---|
| FK506 | 0.8 | 0.99 | 0.71 | 5.78 | 178.4 | 3 | 11 |
| 05-011 | 0.7 | 0.76 | 0.52 | 5.13 | 204.4 | 4 | 12 |
| 05-012 | 0.4 | 0.36 | 0.03 | 4.86 | 215.7 | 4 | 13 |
| 05-013 | 1.4 | 0.93 | 0.61 | 6.02 | 181.6 | 3 | 12 |
| 05-020 | 1.5 | 0.63 | 0.03 | 4.84 | 215.7 | 4 | 13 |
| 05-026 | 3.7 | 0.92 | 1.07 | 5.24 | 219.8 | 4 | 13 |
| 05-027 | 1.1 | 0.89 | 0.03 | 6.36 | 216.7 | 4 | 12 |
| 05-028 | 2.5 | 0.94 | 0.02 | 5.13 | 231.9 | 5 | 14 |
| 05-037 | 8.7 | ND | 0.03 | 3.06 | 241.7 | 5 | 14 |
| 05-050 | 3.6 | ND | 0.70 | 5.88 | 190.7 | 3 | 12 |
| 05-051 | 1.2 | ND | 0.65 | 5.88 | 190.7 | 3 | 12 |
| 05-060 | 0.6 | ND | 0.09 | 4.00 | 213.1 | 3 | 12 |
| 05-061 | 0.2 | ND | 0.01 | 4.00 | 213.1 | 3 | 12 |
| 05-064 | 0.2 | ND | 0.60 | 4.70 | 219.8 | 4 | 13 |
| 05-084 | 3.3 | ND | ND | 6.18 | 190.7 | 3 | 12 |
| 05-085 | 1.5 | ND | 0.56 | 6.18 | 190.7 | 3 | 12 |
| 05-086 | 1.6 | ND | 0.44 | 4.30 | 213.1 | 3 | 12 |
| 05-092 | 2.5 | ND | 1.02 | 4.30 | 213.1 | 3 | 12 |

Phospho-S6 level (fraction of DMSO)
0 — No Blocking    1 — Full Blocking

**Extended Data Fig. 2 | Structures polar FKBP12 ligands synthesized.**
(**a**, SLF derivatives; **b**, FK506 derivatives). Listed in the table are their affinity to recombinant FKBP12 (fluorescence polarization assay) and their efficacy of blocking mTOR inhibition by rapamycin or RapaLink-1 (assessed by western blot analysis of p-S6 level after treatment of MCF7 cells with a combination of 10 nM Rapamycin/Rapalink + 10 μM candidate compound for 24 h. P-S6 level is quantified as fraction of DMSO control). ND, not determined. Calculated logP (cLogP) and topological polar surface area (tPSA) values were calculated using PerkinElmer ChemDraw 16.0. Hydrogen bond donor (HBD) and hydrogen bond acceptor (HBA) numbers were determined at pH 7.4 using ChemAxon Marvin Sketch 20.21. Compound 06–039 is identical to compound 05–084 but prepared in a separate batch. Compound 06–041 is identical to compound 05–086 but prepared in a separate batch.

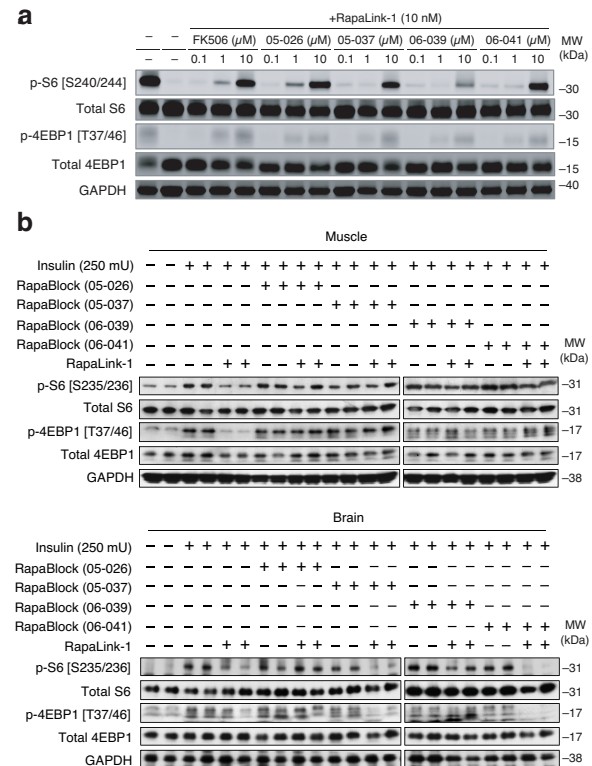

**Extended Data Fig. 3 | Four candidate RapaBlock molecules were evaluated in cells (a) and *in vivo* (b).** GAPDH is a loading control. Data in (a) is representative of two independent experiments. Data in (b) is from one experiment with *n = 2* for each treatment condition. For gel source data, see Supplementary Fig. 1.

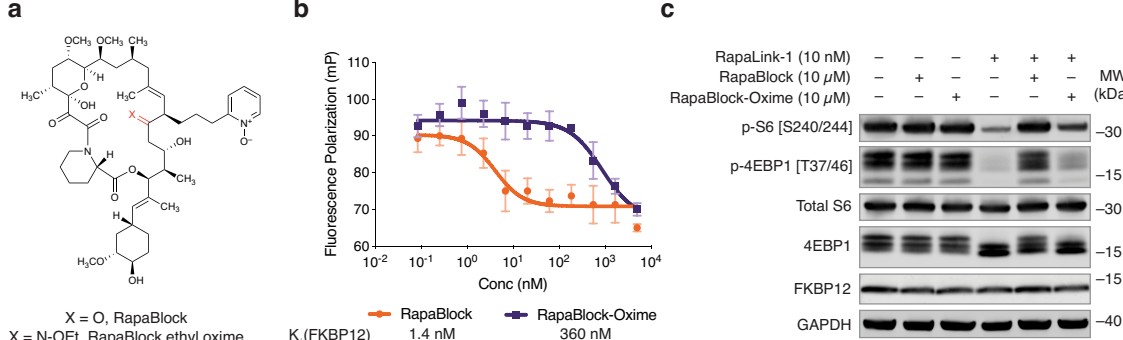

**a**

X = O, RapaBlock
X = N-OEt, RapaBlock ethyl oxime

**b**

$K_i$(FKBP12)

| RapaBlock | RapaBlock-Oxime |
|-----------|-----------------|
| 1.4 nM | 360 nM |

**c**

**Extended Data Fig. 4 | A RapaBlock analog with impaired binding to FKBP12 failed to block RapaLink-1 in cells.** a) Structures of RapaBlock and RapaBlock ethyl oxime. b) RapaBlock ethyl oxime shows attenuated affinity to FKBP12 ($n$ = 3, data are presented as mean values +/− SD). c) RapaBlock ethyl oxime is less potent at blocking RapaLink-1. Data is representative of two independent experiments. For gel source data, see Supplementary Fig. 1.

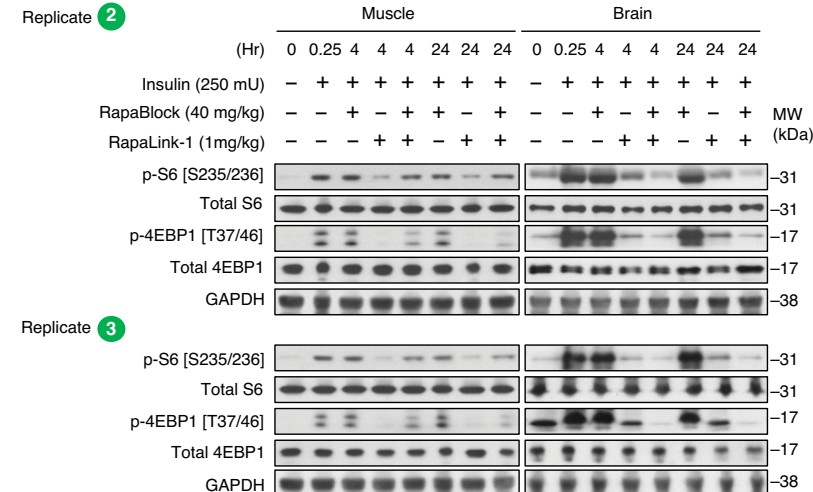

**Extended Data Fig. 5 | Biological replicates of the samples used in the quantification shown in Fig. 4a.** Mice were treated intraperitoneally with a single dose of RapaLink-1 (1 mg/kg), RapaBlock (40 mg/kg), or a combination of both. mTOR activity was stimulated with insulin (250 mU) 15 min before tissue dissection. Whole brain and skeletal muscle were analyzed by immunoblot. The intensities of p-RPS6 and p-4EBP1 were quantified by densitometry using Silver Fast Scanner and ImageJ software and shown below each immunoblot as fold increase relative to muscle or brain without insulin stimulation, normalized to GAPDH. For gel source data, see Supplementary Fig. 1.

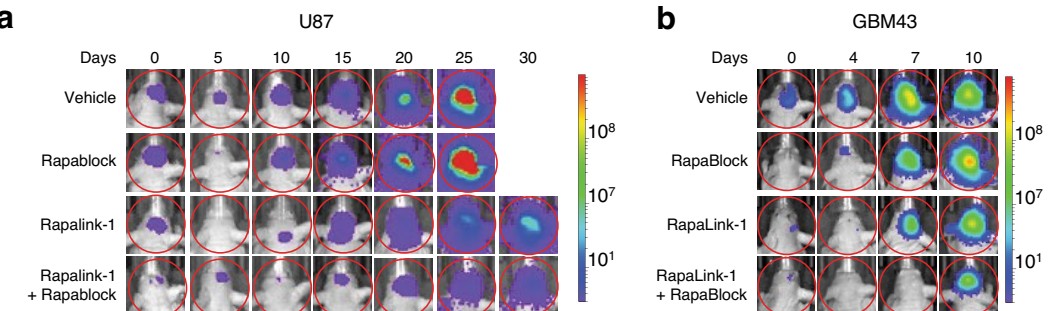

**Extended Data Fig. 6 | Exemplary bioluminescence images of mice bearing intracranial glioblastoma xenografts.** a, U87MG cells expressing firefly luciferase were injected intracranially into BALB/c^nu/nu mice. After tumor establishment, mice were sorted into four groups and treated by i.p. injections every 5 days of vehicle, RapaLink-1 (1 mg/kg), RapaBlock (40 mg/kg), or a combination of RapaLink-1 and RapaBlock (1 mg/kg and 40 mg/kg, respectively). Bioluminescence imaging of tumor-bearing mice was obtained at days shown (day 0 was start of treatment), using identical imaging conditions. b, GBM43 cells expressing firefly luciferase were injected intracranially into BALB/c^nu/nu mice. After tumor establishment, mice were sorted into four groups and treated by i.p. injections every 5 days of vehicle, RapaLink-1 (1.2 mg/kg), RapaBlock (60 mg/kg), or a combination of RapaLink-1 and RapaBlock (1.2 mg/kg and 60 mg/kg, respectively). Bioluminescence imaging of tumor-bearing mice was obtained at days shown (day 0 was start of treatment), using identical imaging conditions.

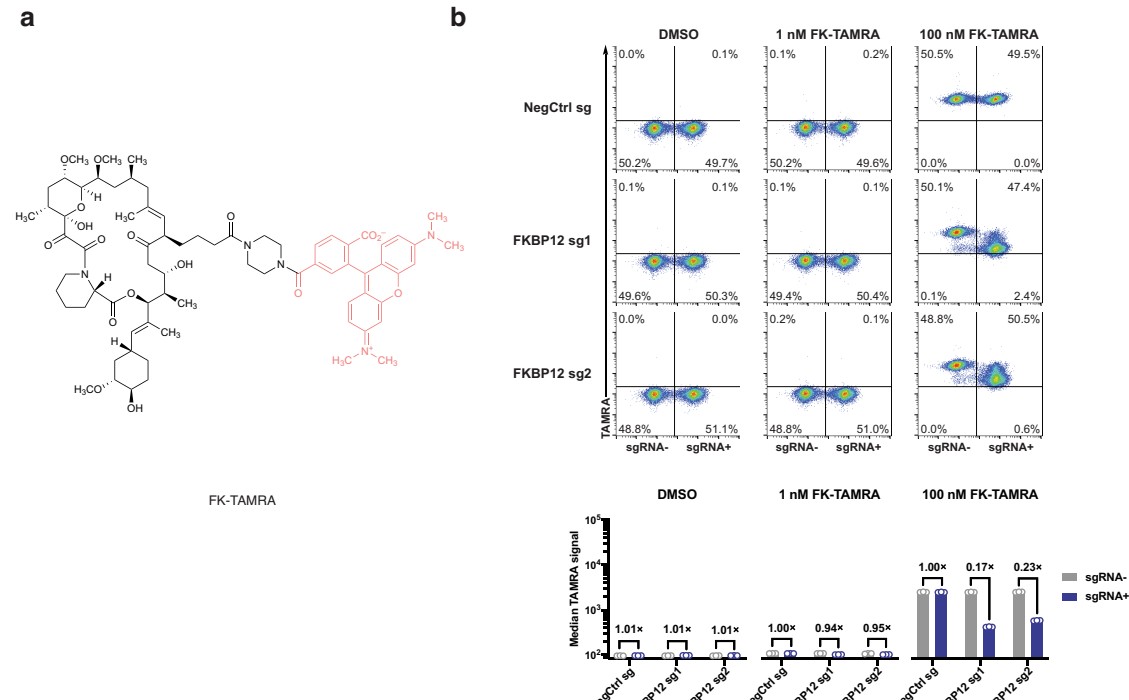

**a**

FK-TAMRA

**Extended Data Fig. 7 | (Related to Extended Data Fig. 1) FKBP12-FK506 interaction contributes to cellular accumulation of a fluorescence analog of FK506. a**, Structure of a fluorescent probe FK-TAMRA. **b**, FKBP12 knockdown decreases cellular retention of FK-TAMRA. Flow cytometry scatter plots shown are representative of three independent replicates, which are summarized in the bar graphs. Data are presented as mean values +/− SD with individual data points. Fold change is calculated as the ratio between means (sgRNA+/sgRNA−). See Supplementary Fig. 2 for gating strategy.

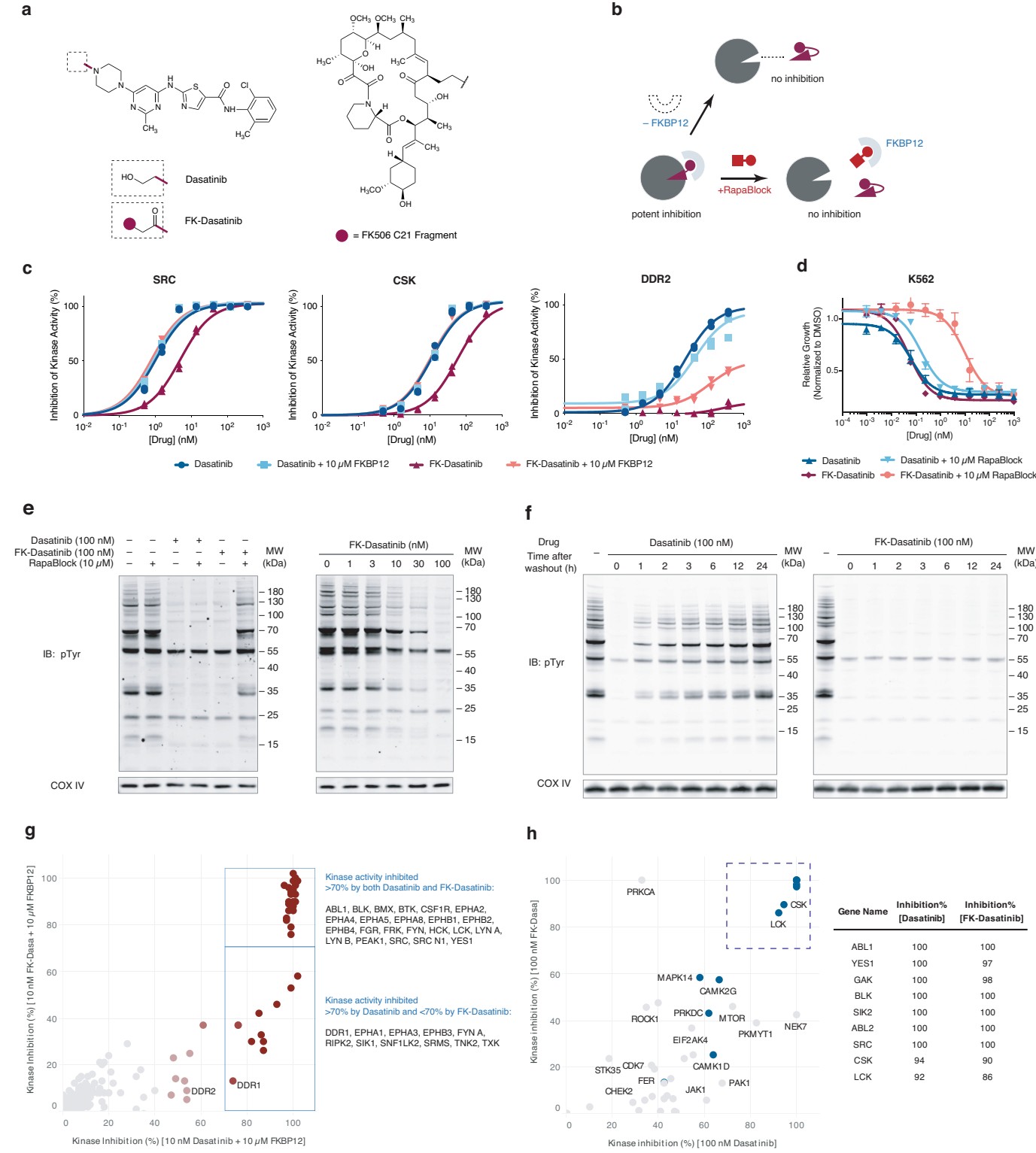

**Extended Data Fig. 8 | FK-Dasatinib is a FKBP12-dependent Src family kinase inhibitor with long cellular retention time. a**, Structures of Dasatinib and FK-Dasatinib. **b**, Proposed working model for FKBP-dependent kinase inhibitors. **c**, Inhibition of Src, Csk and DDR2 kinases by Dasatinib and FK-Dasatinib in the absence or presence of supplemented 10 μM recombinant FKBP12 protein (*n* = 2, data are presented as individual points). **d**, Inhibition of K562 cell proliferation by Dasatinib and FK-Dasatinib, in the presence or absence of 10 μM RapaBlock (*n* = 3, data are presented as mean values +/− SD). **e**, Jurkat cells were stimulated with anti-CD3 antibody (OKT3) in the presence of dasatinib, FK-dasatinib, and/or RapaBlock and analyzed by immunoblot. COX IV is a loading control. Data is representative of two independent experiments.

**f**, Jurkat cells were pulse-treated with Dasatinib (100 nM) or FK-Dasatinib (100 nM) for 1 h, then drugs were washed out and cells were incubated in drug-free media for various amounts of time, stimulated with anti-CD3 antibody (OKT3) for 5 min and analyzed by immunoblot. Results shown are representative of three independent experiments. COX-IV is a loading control. For gel source data, see Supplementary Fig. 1. **g**, inhibition of 476 purified kinases by Dasatinib (10 nM) or FK-Dasatinib (10 nM) in the presence of 10 μM recombinant FKBP12 protein (*n* = 1). See Supplementary Table 1 for a full list of kinase inhibition data. **h**, kinase profiling of Dasatinib (100 nM Dasatinib) or FK-Dasatinib (100 nM) in Jurkat cells using the covalent occupancy probe XO44. Results shown are representative of two independent experiments.

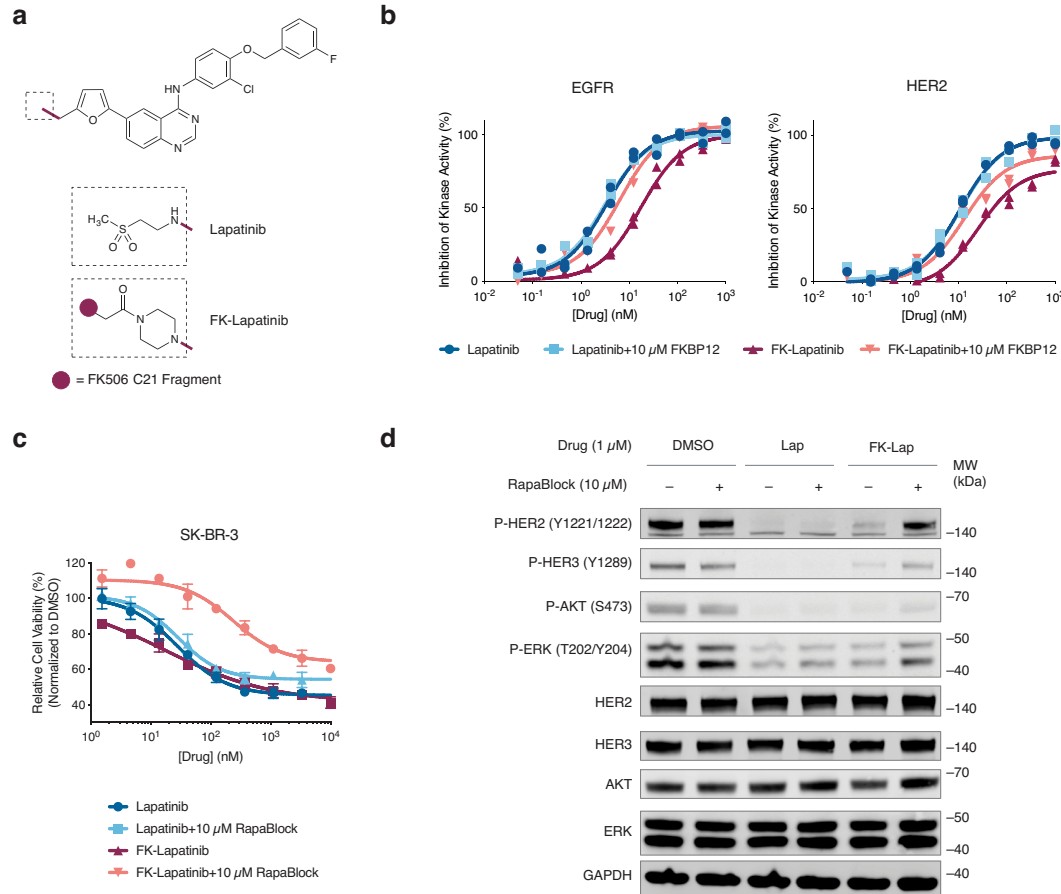

**Extended Data Fig. 9 | FK-Lapatinib is a FKBP12-dependent EGFR/HER2 kinase inhibitor. a**, Chemical structures of Lapatinib and FK-Lapatinib. **b**, Inhibition of EGFR and HER2 kinase activity by Lapatinib and FK-Lapatinib in the absence or presence of supplemented 10 µM recombinant FKBP12 protein ($n = 2$, data are presented as individual points). **c**, Inhibition of proliferation of SK-BR3 cells by lapatinib and FK-lapatinib in the presence of absence of 10 µM RapaBlock ($n = 3$, data are presented as mean values +/− SD). **d**, SK-BR3 cells were treated with Lapatinib, FK-Lapatinib, and/or RapaBlock for 1 h and analyzed by immunoblot. Results shown are representative of three independent experiments. GAPDH is a sample processing control. For gel source data, see Supplementary Fig. 1.

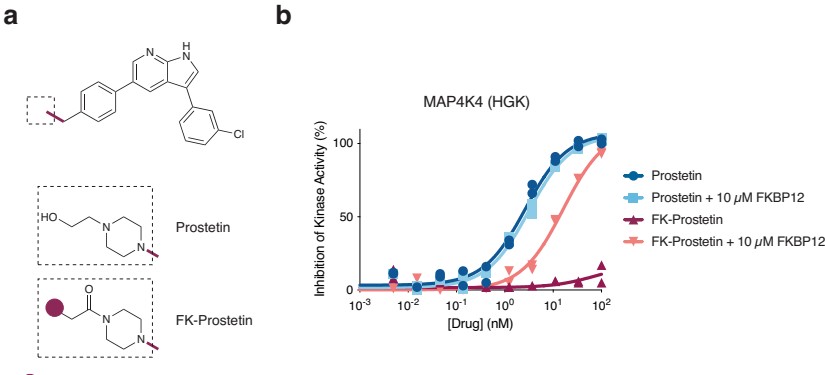

**a**

Prostetin

FK-Prostetin

● = FK506 C21 Fragment

**b**

MAP4K4 (HGK)

- Prostetin
- Prostetin + 10 μM FKBP12
- FK-Prostetin
- FK-Prostetin + 10 μM FKBP12

**Extended Data Fig. 10 | FK-Prostetin is a FKBP12-dependent MAP4K4(HGK) kinase inhibitor. a**, Chemical structures of Prostetin and FK-Prostetin. **b**, Inhibition of MAP4K4(HGK) kinase activity by Lapatinib and FK-Lapatinib in the absence or presence of supplemented 10 μM recombinant FKBP12 protein ($n$ = 2, data are presented as individual points).

# Reporting Summary

Nature Research wishes to improve the reproducibility of the work that we publish. This form provides structure for consistency and transparency in reporting. For further information on Nature Research policies, see our Editorial Policies and the Editorial Policy Checklist.

## Statistics

For all statistical analyses, confirm that the following items are present in the figure legend, table legend, main text, or Methods section.

| n/a | Confirmed | |
|---|---|---|
| ☐ | ☒ | The exact sample size ($n$) for each experimental group/condition, given as a discrete number and unit of measurement |
| ☐ | ☒ | A statement on whether measurements were taken from distinct samples or whether the same sample was measured repeatedly |
| ☐ | ☒ | The statistical test(s) used AND whether they are one- or two-sided *Only common tests should be described solely by name; describe more complex techniques in the Methods section.* |
| ☒ | ☐ | A description of all covariates tested |
| ☒ | ☐ | A description of any assumptions or corrections, such as tests of normality and adjustment for multiple comparisons |
| ☐ | ☒ | A full description of the statistical parameters including central tendency (e.g. means) or other basic estimates (e.g. regression coefficient) AND variation (e.g. standard deviation) or associated estimates of uncertainty (e.g. confidence intervals) |
| ☐ | ☒ | For null hypothesis testing, the test statistic (e.g. $F$, $t$, $r$) with confidence intervals, effect sizes, degrees of freedom and $P$ value noted *Give P values as exact values whenever suitable.* |
| ☒ | ☐ | For Bayesian analysis, information on the choice of priors and Markov chain Monte Carlo settings |
| ☒ | ☐ | For hierarchical and complex designs, identification of the appropriate level for tests and full reporting of outcomes |
| ☒ | ☐ | Estimates of effect sizes (e.g. Cohen's $d$, Pearson's $r$), indicating how they were calculated |

*Our web collection on statistics for biologists contains articles on many of the points above.*

## Software and code

Policy information about availability of computer code

| Data collection | Software used for data acquisition are listed below. Flow cytometry: Thermo Fisher Scientific Attune NxT Software, 3.1.1243.0; Plate reader assays: TECAN SparkControl 2.1; Western Blot: Li-COR Odyssey 2.1.15 Western Blot (film based): SilverFast scan software (Version 6.4.2r9) |
|---|---|
| Data analysis | Flow cytometry: FlowJo 10.7.1 Plate reader assays: Prism 9.0 Western Blot: ImageJ 2.0.0-rc-69/1.52p Calculation of cLogP and tPSA: ChemDraw Professional 16.0 Calculation of hydrogen bond donors and acceptors: MarvinSketch 20.21.0 |

For manuscripts utilizing custom algorithms or software that are central to the research but not yet described in published literature, software must be made available to editors and reviewers. We strongly encourage code deposition in a community repository (e.g. GitHub). See the Nature Research guidelines for submitting code & software for further information.

## Data

Policy information about availability of data

 All manuscripts must include a data availability statement. This statement should provide the following information, where applicable:
- Accession codes, unique identifiers, or web links for publicly available datasets
- A list of figures that have associated raw data
- A description of any restrictions on data availability

Unprocessed gel images for all immunoblots (figures 1d, 3a, 3d, 4a, extended data figures 3, 4c, 5, 7e, 7f, 8d) are provided in Supplementary Figure 1. Raw data for the kinase inhibition assays, cell growth assays and xenograft studies are available as source data files. All data that support the findings of this study are available from the corresponding author upon reasonable request.

Response to editor: The plate reader assay data in this manuscript are very small data sets using specialized assays (all described in the manuscript) which are not suitable for deposition in public data repositories. The manuscript does not contain new nucleic acid or protein structures. We are more than happy to provide the raw data upon request to the corresponding author.

# Field-specific reporting

Please select the one below that is the best fit for your research. If you are not sure, read the appropriate sections before making your selection.

☒ Life sciences          ☐ Behavioural & social sciences          ☐ Ecological, evolutionary & environmental sciences

For a reference copy of the document with all sections, see nature.com/documents/nr-reporting-summary-flat.pdf

# Life sciences study design

All studies must disclose on these points even when the disclosure is negative.

| | |
|---|---|
| Sample size | Except for the kinase activity assays performed by thrid-party services, all in vitro experiments were performed in at least three replicates. For western blot analyses of mTOR signaling, three mice per treatment group were used. For U87MG xenograft studies, seven mice per treatment were used. For GBM43 xenograft studies, five mice per treatment were used. These numbers represent the maximum reasonable sample size that allowstimely administration of drugs and processing of tissue samples. |
| Data exclusions | No data was excluded from the analysis. |
| Replication | Kinase activity assays performed by third-party services (Figs 1b and 5c, Extended Figs 6c, 7b and 8b) were performed once with two technical replicates. Animal experiments were performed once with specified numbers of animals. All other experimental findings have been independently replicated at least twice (see figure captions for the number of replicates for individual experiments). |
| Randomization | For western blot analysis of mTOR signaling, mice were randomized into treatment groups. For U87MG tumor xenograft experiments, mice were sorted into four groups of equal mean bioluminescent signal at the beginning of the treatment(7 mice per group). For GBM43 tumor xenograft experiments, mice were sorted into four groups of equal mean bioluminescent signal at the beginning of the treatment(5 mice per group). Each biochemical experiment in this study is rationally designed and leads to a specific conclusion. Samples were not randomized for these experiments. |
| Blinding | Researchers were blinded from mice group assignment for imaging luciferase activity and measuring tumor burden. Solutions of RapaBlock were significantly more viscous than vehicle and solutions of RapaLink-1, and therefore full blinding in drug treatment was not attainable. Due to the short duration of treatment and limitation of personnel, animal experiments that analyze acute changes in mTOR signaling (Fig 4a and Extended Fig 3b) in this study were not blinded. Each biochemical experiment in this study is rationally designed and leads to a specific conclusion. Samples were not blinded for these experiments. |

# Reporting for specific materials, systems and methods

We require information from authors about some types of materials, experimental systems and methods used in many studies. Here, indicate whether each material, system or method listed is relevant to your study. If you are not sure if a list item applies to your research, read the appropriate section before selecting a response.

## Materials & experimental systems

| n/a | Involved in the study |
|---|---|
| ☐ | ☒ Antibodies |
| ☐ | ☒ Eukaryotic cell lines |
| ☒ | ☐ Palaeontology and archaeology |
| ☐ | ☒ Animals and other organisms |
| ☒ | ☐ Human research participants |
| ☒ | ☐ Clinical data |
| ☒ | ☐ Dual use research of concern |

## Methods

| n/a | Involved in the study |
|---|---|
| ☒ | ☐ ChIP-seq |
| ☐ | ☒ Flow cytometry |
| ☒ | ☐ MRI-based neuroimaging |

## Antibodies

| Antibodies used | A full list of antibodies used in provided in Supplementary Information. |
|---|---|
| Validation | P-Akt[S473] antibody (CST-4060) was validated by the manufacturer using wortmannin-treated and PDGF-treated samples as negative and positive controls, respectively. We independently validated this antibody using a pan-mTOR inhibitor MLN0128 in our experiments. https://www.cellsignal.com/products/primary-antibodies/phospho-akt-ser473-d9e-xp-rabbit-mab/4060 |

Akt antibody (CST-2920) was validated by the manufacturer using recombinant GST proteins and standard human cell lines. https://media.cellsignal.com/coa/2920/8/2920-lot-8-coa.pdf

P-S6 (S240/244) antibody (CST-5364) was validated by the manufacturer using insulin- or FBS-treated samples. We independently validated this antibody using the mTORC1 inhibitor rapamycin in our experiments. https://www.cellsignal.com/products/primary-antibodies/phospho-s6-ribosomal-protein-ser240-244-d68f8-xp-rabbit-mab/5364

P-S6 (S235/236) antibody (CST-4858) was validated by the manufacturer using lambda-phosphatase- and FBS-treated samples. We independently validated this antibody using the mTORC1 inhibitor rapamycin in our experiments. https://www.cellsignal.com/products/primary-antibodies/phospho-s6-ribosomal-protein-ser235-236-d57-2-2e-xp-rabbit-mab/4858

S6 antibody (CST-2217) was validated by the manufacturer using standard human and mouse cell lines and has been cited by 1130 publications. https://www.cellsignal.com/products/primary-antibodies/s6-ribosomal-protein-5g10-rabbit-mab/2217

P-4EBP1[T37/46] antibody (CST-2855) was validated by the manufacturer using insulin-treated samples. We independently validated this antibody using a pan-mTOR inhibitor MLN0128 in our experiments. https://www.cellsignal.com/products/primary-antibodies/phospho-4e-bp1-thr37-46-236b4-rabbit-mab/2855

4EBP1 antibody (CST-9644) was validated by the manufacturer using a cell line that contains mutation in the 4EBP1 gene that causes the production of a higher-molecular weight protein. https://www.cellsignal.com/products/primary-antibodies/4e-bp1-53h11-rabbit-mab/9644

FKBP12 antibody (abcam58072) was validated using recombinant FKBP12 proteins produced in-house.

Actin antibody (Proteintech, 60008-1-Ig) was validated by the manufacturer using a range of cell lines in Western blot and immunofluorescence, and has been cited by 1972 publiactions. https://www.ptglab.com/products/ACTB-Antibody-60008-1-Ig.htm

GAPDH antibody (Proteintech, 60004-1-Ig) was validated by the manufacturer using a range of cell lines in Western blot and immunofluorescence, and has been cited by 2971 publiactions. https://www.ptglab.com/products/GAPDH-Antibody-60004-1-Ig.htm

P-Tyr(4G10) antibody (EMD Millipore, 05-321) was validated by the manufacturer using EGF-treated A431 cell lysates. https://www.emdmillipore.com/US/en/product/Anti-Phosphotyrosine-Antibody-clone-4G10,MM_NF-05-321

COX IV antibody (CST-4850) was validated by the manufacturer using a range of cell lines. https://www.cellsignal.com/products/primary-antibodies/cox-iv-3e11-rabbit-mab/4850

P-ERK[T202/Y204] antibody (CST-9101) was validated by the manufacturer using recombinant phospho-MAPK proteins. https://www.cellsignal.com/products/primary-antibodies/phospho-p44-42-mapk-erk1-2-thr202-tyr204-antibody/9101

ERK antibody (CST-4695) was validated by the manufacturer using specific siRNA knockdowns and has been cited by 2827 publications. https://www.cellsignal.com/products/primary-antibodies/p44-42-mapk-erk1-2-137f5-rabbit-mab/4695

P-HER2[Y1221/1222] antibody (CST-2243) was validated by the manufacturer using EGF-stimulated cell lysates. https://www.cellsignal.com/products/primary-antibodies/phospho-her2-erbb2-tyr1221-1222-6b12-rabbit-mab/2243

P-HER3[Y1289] antibody (CST-2842) was validated by the manufacturer using human neuregulin-1-stimulated cell lysates. https://www.cellsignal.com/products/primary-antibodies/phospho-her3-erbb3-tyr1289-d1b5-rabbit-mab/2842

HER2 antibody (CST-4290) was validated by the manufacturer using authentic cell lines with known HER2 expression profile. https://www.cellsignal.com/products/primary-antibodies/her2-erbb2-d8f12-xp-rabbit-mab/4290

HER3 antibodies (CST-4754) was validated by the manufacturer using authentic cell lines with known HER3 expression profile. https://www.cellsignal.com/products/primary-antibodies/her3-erbb3-1b2e-rabbit-mab/4754

Goat anti-rabbit IgG-IRDye 800 was validated by the manufacturer: https://www.licor.com/documents/rfm2hw40wf33p06f3ndjrcorwi5usbft

Goat anti-mouse IgG-IRDye 680 was validated by the manufacturer: https://www.licor.com/documents/7bohf1sfzugccz22fh0um00cvz8ocizf

## Eukaryotic cell lines

Policy information about cell lines

| Cell line source(s) | MCF7 cells were obtained from ATCC (HTB-22) and maintained in 1:1 DMEM:F12 (Gibco) + 10% heat-inactivated fetal bovine serum (FBS, Axenia Biologix) supplemented with 4 mM L-glutamine, 100 U/mL penicillin and 100 U/mL streptomycin (Gibco). SK-BR-3 cells were obtained from ATCC (HTB-30) and maintained in McCoy's 5A (Gibco) + 10% heat-inactivated FBS supplemented with 2 mM L-glutamine, 100 U/mL penicillin and 100 U/mL streptomycin (Gibco). K562 CRISPRi cells were a |
|---|---|

gift from Dr. Luke Gilbert and maintained in RPMI 1640 (Gibco) + 10% heat-inactivated FBS supplemented with 2 mM L-glutamine, 100 U/mL penicillin, 100 U/mL streptomycin (Gibco) and 0.1% Pluronic F-68 (Gibco). RAW264.7 cells were obtained from ATCC (TIB-71) and maintained in DMEM (Gibco) + 10% heat-inactivated FBS supplemented with 2 mM L-glutamine, 100 U/mL penicillin and 100 U/mL streptomycin (Gibco). Jurkat cells were obtained from ATCC (TIB-152) and maintained in RPMI 1640 (Gibco) + 10% heat-inactivated FBS supplemented with 2 mM L-glutamine, 100 U/mL penicillin and 100 U/mL streptomycin (Gibco). Jurkat-Lucia NFAT cells were obtained from InvivoGen and maintained in IMDM (Gibco) + 10% heat-inactivated FBS supplemented with 2 mM L-glutamine, 100 U/mL penicillin, 100 U/mL streptomycin (Gibco) and 100 µg/mL Zeocin (InvivoGen). HEK293T cells were obtained from UCSF Cell Culture Facility and maintained in 1:1 DMEM:F12 (Gibco) + 10% heat-inactivated fetal bovine serum (FBS, Axenia Biologix) supplemented with 4 mM L-glutamine, 100 U/mL penicillin and 100 U/mL streptomycin (Gibco).

| | |
|---|---|
| Authentication | Cell lines were obtained from vendors noted above and all cell experiments were performed within 15 passages from the initial ATCC stock. Cell morphology were regularly inspected. Cell lines from ATCC were STR profiled by the manufacturer. Jurkat-Lucia NFAT cells is a commercial cell line developed by InvivoGen and was authenticated by the manufacturer. Cell lines from UCSF Cell Culture Facility were STR profiled by the UCSF Cell Culture Facility. No further authentications were performed after the purchase. |
| Mycoplasma contamination | All cell lines were tested mycoplasma negative using MycoAlert™ Mycoplasma Detection Kit (Lonza). |
| Commonly misidentified lines (See ICLAC register) | No commonly misidentified cell lines were used in the study. |

# Animals and other organisms

Policy information about studies involving animals; ARRIVE guidelines recommended for reporting animal research

| | |
|---|---|
| Laboratory animals | 4-6 week-old female athymic BALB/Cnu/nu athymic nude mice were used in vivo experiments. All mice at the Helen Diller Cancer center UCSF are housed in individually ventilated microisolator cages. The housing racks have automatic water that is sterilized through filtration, reverse osmosis, and UV light exposure. All mice are fed a standard diet that is pre-irradiated. The light/dark cycle is 12 light/12 dark. Temperature is maintained in the range of 68-74 degrees Fahrenheit. Humidity is maintained at 30-70%. |
| Wild animals | No wild animals were used in the study. |
| Field-collected samples | No field-collected samples were used in the study. |
| Ethics oversight | All animal experiments were conducted using protocols approved by University of California, San Francisco's Institutional Animal Care and Use Committee (IACUC). |

Note that full information on the approval of the study protocol must also be provided in the manuscript.

# Flow Cytometry

## Plots

Confirm that:

☒ The axis labels state the marker and fluorochrome used (e.g. CD4-FITC).

☒ The axis scales are clearly visible. Include numbers along axes only for bottom left plot of group (a 'group' is an analysis of identical markers).

☒ All plots are contour plots with outliers or pseudocolor plots.

☒ A numerical value for number of cells or percentage (with statistics) is provided.

## Methodology

| | |
|---|---|
| Sample preparation | Untransduced (No virus or sgRNA-) and sgRNA expression vector-transduced (sgRNA+) K562 CRISPRi cells were mixed in a 1:1 ratio, plated in 48-well plates, and incubated overnight. Cell mixtures were treated with the indicated TAMRA-linked compounds (300 µL final volume) and incubated for 24 h. Plates were placed over ice and 200 µL of each compound-cell mixture was transferred to a 96-well U-bottom plate. Cells were pelleted at 500g for 5 min, and subsequently washed 2× with ice-cold FACS buffer (PBS + 1% BSA + 0.1% NaN3). Cells were resuspended in 200 µL of ice-cold FACS buffer and assessed using an Attune NxT (Thermo Fisher Scientific). Relative uptake of TAMRA-linked compounds was determined by comparing TAMRA fluorescence (561 nm excitation laser, 585/16 emission filter) between sgRNA- and sgRNA+ cells within each well. |
| Instrument | Thermo Fisher Scientific Attune NxT |
| Software | FlowJo 10.7.1 |
| Cell population abundance | 50 µL of cell mixture was assayed (typically ≥ 10,000 total events) on an Attune NxT (Thermo Fisher Scientific). The gating strategy proceeded as follows: 1) A forward scatter area (FSC-A) vs. side scatter area (SSC-A) gate was used to exclude debris (~90% gated), 2) A FSC-A vs. forward scatter height (FSC-H) gate was used to exclude doublets (~90% gated), 3) A VL1 height (VL1-H, 405 nm excitation laser, 440/50 emission filter) vs. YL1 height (YL-H, 561 nm excitation laser, 585/16 emission filter) gate was used to assess sgRNA expression and TAMRA-linked compound uptake respectively |

Gating strategy

Gating strategy is provided in Supplemental Fig. 2. This gating strategy is used for data shown in Extended Data Figure 1c and 6b.
Response to editor: this gating strategy is a graphical demonstration and cannot be directly placed in the reporting summary.

☒ Tick this box to confirm that a figure exemplifying the gating strategy is provided in the Supplementary Information.

