## [Peer Review File · Nature]

Manuscript Title: Brain-Restricted mTOR Inhibition with Binary Pharmacology

Reviewer Comments & Author Rebuttals

Reviewer Reports on the Initial Version:

Referees' comments:

Referee #1 (Remarks to the Author):

The study by Zhang et al describes a new approach to modify existent drugs to maintain their on-target activity in the brain but shed their on-target activity in peripheral tissues. The work builds on the development by the same authors of bivalent mTOR inhibitors that exploit the juxtaposition of two drug-binding pockets (FKBP12–rapamycin-binding domain and the kinase domain)(PMID: 27279227). In their current work, they propose that FKBP12-dependent mTOR inhibition (through rapamycin or their bivalent third generation mTOR inhibitor RapaLink-11) can be limited to the brain by simultaneous administration of a novel brain-impermeable FKBP12 ligand (so-called “RapaBlock”). They also show that this approach of achieving FKBP12-dependent kinase activity could be engineered into other ATP-competitive kinase inhibitor. The work is conceptually novel, but there are gaps in the presented data that make it difficult to fully assess the importance and potential impact of this work.

Main critiques:

(1.) Key objectives in designing “brain-selective” pharmacological approaches include the elimination of systemic toxicities at currently used dosing schedules, the ability to explore higher doses by widening the therapeutic window of these drugs, or a combination of both. Unfortunately, neither of these questions are fully addressed experimentally. For example, the authors claim that combination of Rapamycin or RapaLink-1 with Rapablock reduces the well-documented immunosuppression associated with rapalogs, but the data to support this claim is only generated ex-vivo (by examining effects on anti-CD3/anti-CD28 induced PBMC expansion). The authors also do not show that the combination with Rapamycin or RapaLink-1 with Rapablock allows them to consider higher (and perhaps more effective) doses of Rapamycin or Rapalink-1 which would result in dose-limiting toxicities in the absence of Rapablock. The absence of these types of in-vivo experiments for RapaLink-1 or any of other FK506-conjugated kinase inhibitors makes it difficult to fully evaluate the overall premise of the work that blockade of FKBP12 in peripheral tissues represents a biologically/therapeutically meaningful advance.

(2.) The rationale to focus on mTOR inhibition in GBM is not sufficiently clear. There is no data to support the claim that “mTOR inhibitors have shown efficacy in GBM” (page 3). In fact, none of the (multiple) clinical trials with rapalogs or mTOR kinase inhibitors have shown clinical activity in GBM and it is far from clear whether this failure is due to inadequate mTOR kinase inhibition (versus feedback-reactivation of AKT, mTOR independent signaling pathways, other). The U87 GBM cells line used in the current study is known to be exceptionally responsive to mTOR inhibitors, but has not been predictive for the experience with mTOR inhibitors in GBM patients. More extensive in-vivo studies using multiple different orthotopic GBM PDX models, which faithfully represent the invasive

growth and genetics of human GBM, are required to show that: (1.) mTOR inhibition has antitumor activity which is closely linked to the degree of mTOR inhibition in tumor cells, and that (2) the combination of Rapamycin or RapaLink-1 with RapaBlock is at least some ways “better” (more antitumor activity or less systemic toxicity) than Rapamycin or RapaLink-1 in the absence of RapaBlock. Show that the addition of Rapablock to RapaLink-1 does not reduce the antitumor activity of RapaLink-1 is important, but not compelling enough.

Referee #2 (Remarks to the Author):

This is an exciting study that exploits the surprising finding that effective cellular engagement of kinases by bivalent molecules composed of an ATP – competitive kinase inhibitor linked to FKBP12 binders requires successful engagement of FKBP12. The authors show that this activity can be exploited to achieve brain selective pharmacology by blocking effective engagement in tissues with a FKBP12 blocking agent. Impressively the authors demonstrate proof of concept for four different kinases showing the potential generality for the approach. Overall, I find this to be a very exciting study and the conclusions are well supported by the presented data and I think it is meritorious of publication. A couple of minor points to address:

1. The authors convincingly demonstrate the requirement for FKBP12 engagement to achieve cellular potency through FKBP12 knock-down and chemical competition experiments. Another useful control would be to show a negative control compound containing a subtle chemical change to the FKBP12 binder that reduced FKBP12 binding and examining its effect on potency. I would assume the authors have prepared such a compound for one of their test systems.
2. I didn't see any pharmacokinetic data for the rapablock compound. It would be good to show PK/PD relationship in peripheral tissues to establish that drug exposure in the periphery by rapablock is indeed required to suppress the pharmacology of the Rapa-link mTOR inhibitor.
3. I didn't understand why the authors pursued macrocyclic rapamycin/FK506-kinase inhibitor hybrids rather than the smaller and synthetically more tractable SLF type-kinase inhibitor hybrids. Perhaps this could be explained more clearly in the text.
4. The authors should show that rapablock is pharmacologically silent (cell proliferation, RNA sequencing etc) if this agent is really just supposed to be a blocking agent.

Referee #3 (Remarks to the Author):

Summary of the key results

Zhang et. al. introduce a novel strategy to achieve brain-specific inhibition of mTOR, while sparing mTOR activity systemically. The authors use a brain-permeable mTOR inhibitor RapaLink-1 and brain impermeable FKBP12 ligand dubbed RapaBlock. The approach is designed to mitigate systemic side effects of mTOR inhibitors when targeting brain cancer or neurological disorders such as TSC. The approach might increase the therapeutic window of FKBP12-binding drugs for the treatment of central nervous system diseases, such as RapaLink-1.

Originality and significance

The approach is novel and has landmark character

Data & methodology: validity of approach, quality of data, quality of presentation

The presentation and interpretation need improvement (see below).

The methodology is sound, the description and (chemical) characterization is lacking in part (see below)

Appropriate use of statistics and treatment of uncertainties

Number of experiments and quantifications need improvement (see below)

Conclusions: robustness, validity, reliability

The major conclusions are robust, some parts might be refined and put into a broader context (see below).

References: appropriate credit to previous work?

Some suggestions below.

Clarity and context: lucidity of abstract/summary, appropriateness of abstract, introduction and conclusions

Minor Revisions needed.

Suggested improvements: experiments, data for possible revision

Major points:

1) Introduction: the sentence "Although mTOR inhibitors have shown efficacy in treating a number of central nervous system (CNS) diseases including tuberous sclerosis complex (TSC)^{3,4}, glioblastoma (GBM)^{5,6} [...]" needs clarification. Are the authors referring only to allosteric inhibitors (rapamycin/rapalogs) or also to ATP-competitive inhibitors? In addition to allosteric mTORC1 inhibitors, ATP-competitive molecules are currently investigated in preclinical studies for the treatment of Tuberous Sclerosis complex (for example <Rageot, 2018, 30359003><Bonazzi, 2020, 31955578><Borsari, 2020, 33166139>; references indicated as <author, year, PMID>. Regarding glioblastoma, AZD2014 has been recently evaluated in a Phase 1 clinical trials [NCT02619864].

2) Figure 1 (and text line 61ff): the nature of the in vitro kinase assay should be mentioned in the figure legend. The link to Thermofischer does not qualify as "methods" and is not functional. That FKBP12 does not affect the Z'-LYTE assay (?) is not surprising, as small peptides are used as substrates. It seems that the available literature explains the observed data concerning in vitro and cellular results presented here. The authors should put this in a better perspective.

3) Line 77ff. The FRB-Rapa(link)-FKBP12 sandwich is certainly important as a "intracellular sink for RapaLink-1 to accumulate in the cell." Instead of Refs 19,20 biochemical work should here be considered illustrating the "quasi-irreversibility" of the ternary complex. There is literature explaining the effect of rapamycin and FKBP12 blocking S6K, S6, 4EBP, etc. phosphorylation by TORC1 elucidating mechanistical aspects.

4) The above discussion of the "intracellular sink" is also important for the key message of the work,

the CNS action of Rapalink-1: Results concerning pharmacokinetics (PK) in the RapaLink-1 and RapaBlock context are not presented here, and are also not established in <Fan et al., 2017, 28292440> cited for Rapalink-1 brain penetration. It should be clearly distinguished between pharmacodynamics (PD) and PK. Ideally the authors would present plasma and brain levels including “fraction unbound” (fu) values for RapaLink-1 and RapaBlock, integrating these values to better illustrate the physiologic mode of action of the RapaLink-1 and RapaBlock. It is likely that the observed PD is dominated by the very high affinity of RapaLink-1 to TORC1, and not its actual “brain permeability” (which is probably comparable to other rapalogs).

5) A conclusive demonstration that RapaBlock is not immunosuppressive is not really provided, although the authors results demonstrate sparing of TORC1 signaling. The authors should either provide a more differentiated text (not referring to mTOR PD experiments as “non-immunosuppression”, or add real in vivo data comparing RapaLink plus/minus RapaBlock in a relevant immune response.

6) Extended Data Figure 2: a legend on how to apply the color code should be included to improve clarity. The statement “none of the SLF derivatives were found to be effective”? should be further elucidated. SLF derivatives have a lower affinity to recombinant FKBP12 with respect to FK506 analogs, but some of them showed efficacy in blocking mTOR inhibition by rapamycin.

7) Authors should comment on the correlation between blocking of mTOR inhibition by rapamycin and RapaLink-1? Some compounds showed a different behavior (e.g. 03-087). What trend was observed (compounds more effective in blocking rapamycin action) and what is the mechanistic basis?

8) What strategy has guided the selection of the four RapaLink-1 blocking compounds? The authors should clarify the threshold values considered for the selection. Some modelling of equilibria ligand-protein complexes would also be very helpful to demonstrate the action of the RapaLink-1/RapaBlock.

9) In extended data Figure 3, the four molecules are 05-026, 05-037, 06-039 and 06-041. However, 06-039 and 06-041 are not listed in Extended Data Figure 2. The chemical structure and the activity of these two derivatives have to be included in Extended Data Figure 2.

10) Among the four selected RapaLink-1 blocking compounds, it looks like there are no pyridine N-oxide derivatives, that would be 05-060, 05-061, 05-086 and 05-092. Could the authors review the numbering and clarify this point?

11) Please include the compounds’ numbers in the text description to guide the readers through the results: “A pyridine N-oxide derivative of FK506 (XX-XXX)”.

12) In the supporting information, RapaBlock has to be matched with the code of the main text (05-086).

13) Figure 4: the labelling is too small and needs to be increased. The gray color of the vehicle is difficult to see.

14) Same Figure: the quantification for the Western blot showing that “while mTOR activity in the brain was inhibited at a comparable level to mice treated with RapaLink-1 only, mTOR activity in skeletal muscle was not affected” needs to be included. Western blots should be quantified. n=1 and n=2 experiments are problematic for useful (non-parametric) statistics.

15) Figure 5 and associated text. The other FK-drugs are “sunked” into cells, but the overall mode of action is not comparable with the RapaLink approach, where both binding sites are in the same complex (TORC1). In this respect the section is an interesting outlook to “brain-specific” action of target, but the characterization of the FK-compounds is a bit scarce.

16) Extended Data Figure 5: Panel g and h need clarification both in the main text and in the Figure legend. What is “1” standing for? Is “Dasatinib and 1” meaning to “Dasatinib and FK-Dasatinib”? The authors state that identical spectrum of kinase targets was observed for Dasatinib and FK-Dasatinib, with the exception of DDR2. Where could this be observed in Extended Figure 5? Labeling for DDR2 is missing.

Minor points:

17) Fig. 5: choose a color different from gray for the circles (badly visible in the printed version).

Formatting and typos:

18) remove brackets: In “(K562/dCas9-KRAB cells)” \diamond In K562/dCas9-KRAB cells

19) Figure 2 legend: “an FKBP12-dependent mTOR inhibitor” \diamond “a FKBP12-dependent mTOR inhibitor”

20) Increase the resolution of Extended Data Figure 1 (including the resolution of the chemical structure RapaTAMRA).

21) The synthesis and characterization of RapaTAMRA should be included in supporting information.

In Supporting information:

22) Add the synthesis and characterization of RapaTAMRA.

23) Are the product UV active after the addition of the pyridine moiety? Can HPLC be used to determine the purity of the final compounds?

24) S1 and S2 have to be related to the numbering reported in the main text.

25) The IUPAC name or SMILE code should be added for the final compounds reported in SI.

26) The authors state that ¹³C NMR peaks of both rotamers are reported collectively. The number of carbons in S2 is 82 instead of 98 considering the rotamers. Could the authors comment on this? Are there peaks corresponding to different carbons (for example methoxy groups are equal)?

27) For the ¹H NMR tabulation of one rotamer, protons are missing e.g. S2 has 74 protons and 69 are tabulated. If not all the protons can be detected (e.g. the OH), it should be stated in the tabulation.

28) According to Nature guidelines on “Characterization of chemical materials”, the authors have to “provide adequate data to support assignment of identity and purity _for each new compound described_. For most organic and organometallic compounds chemical identity should be established through spectroscopic analysis. Please provide standard peak listings for ¹H-NMR and proton-decoupled ¹³C-NMR for all new compounds, with other NMR data (³¹P-NMR, ¹⁹F-NMR, and so on) when appropriate. For new materials, we require high-resolution mass spectral (HRMS) data to support molecular weight identity.” In case of “Combinatorial compound libraries, descriptions of the preparation of combinatorial libraries should include _standard characterization data for a diverse panel of library members._”

29) The authors need to report the synthesis and characterization of the representative SLF and FK506 derivatives (reported in Extended Data Figure 2).

30) Moreover, the synthesis and full characterization of ATP-site kinase inhibitors linked to FK506 have to be reported in the supporting information since these are new compounds developed as proof-of-concept of the broader feasibility of the proposed strategy. Specifically, FK-GNE7915, FK-Dasatinib and FK-Lapatinib should be included. Since they are UV compounds, addition of HPLC chromatograms (and purity assessment) is strongly recommended.

31) Ref 31 lacks journal

Author Rebuttals to Initial Comments:

Referee comments:

Referee #1 (Remarks to the Author):

The study by Zhang et al describes a new approach to modify existent drugs to maintain their on-target activity in the brain but shed their on-target activity in peripheral tissues. The work builds on the development by the same authors of bivalent mTOR inhibitors that exploit the juxtaposition of two drug-binding pockets (FKBP12–rapamycin-binding domain and the kinase domain)(PMID: 27279227). In their current work, they propose that FKBP12-dependent mTOR inhibition (through rapamycin or their bivalent third generation mTOR inhibitor RapaLink-11) can be limited to the brain by simultaneous administration of a novel brain-impermeable FKBP12 ligand (so-called “RapaBlock”). They also show that this approach of achieving FKBP12-dependent kinase activity could be engineered into other ATP-competitive kinase inhibitor. The work is conceptually novel, but there are gaps in the presented data that make it difficult to fully assess the importance and potential impact of this work.

Main critiques:

(1.) Key objectives in designing “brain-selective” pharmacological approaches include the elimination of systemic toxicities at currently used dosing schedules, the ability to explore higher doses by widening the therapeutic window of these drugs, or a combination of both. Unfortunately, neither of these questions are fully addressed experimentally. For example, the authors claim that combination of Rapamycin or RapaLink-1 with Rapablock reduces the well-documented immunosuppression associated with rapalogs, but the data to support this claim is only generated ex-vivo (by examining effects on anti-CD3/anti-CD28 induced PBMC expansion). The authors also do not show that the combination with Rapamycin or RapaLink-1 with Rapablock allows them to consider higher (and perhaps more effective) doses of Rapamycin or Rapalink-1 which would result in dose-limiting toxicities in the absence of Rapablock. The absence of these types of in-vivo experiments for RapaLink-1 or any of other

FK506-conjugated kinase inhibitors makes it difficult to fully evaluate the overall premise of the work that blockade of FKBP12 in peripheral tissues represents a biologically/therapeutically meaningful advance.

This is an excellent point – we thank the reviewer for pointing this out. We have made the following additions/changes to the manuscript:

1. We have now included additional data and discussion to address how the combination of RapaLink-1 and RapaBlock allows us to achieve chronic mTOR inhibition in the brain *in vivo*. Some of the data which addresses this issue was included in a manuscript originally submitted with this manuscript but which was then published in Ehinger et al. *Nat. Comm. (2021)* entitled, "Brain-specific inhibition of mTORC1 eliminates side effects resulting from mTORC1 blockade in the periphery and reduces alcohol intake in mice." Since much of the toxicity mitigation data was included in that study we reference it in the revision but also include a key figure

above. Over the course of a 4 week dosing experiment in an alcohol use disorder paradigm we saw less body weight loss (b) with the inclusion of RapaBlock in RapaLink-1 treatment, additionally we saw mitigation of hyperglycemia in a glucose tolerance test (c) at the end of the experiment. The Ehinger, et. al. manuscript also addresses other toxicity aspects including liver toxicity and is now referenced in the revised manuscript.

- We agree with the reviewer that the ex vivo PBMC proliferation data does not fully address immune suppression in vivo. We have performed extensive *in vivo* experiments (T cell counting, various T cell, B cell, etc. as well as mass cytometry with over 40 cell surface markers) using Rapamycin as a control, along with RapaLink-1, RapaBlock, and RapaLink-1 + RapaBlock. Frustratingly, we could not obtain convincing positive control data of immune marker suppression or activation of regulatory T cell markers with Rapamycin in a robust manner. Reading through the Rapamycin immune suppression literature it became clear that there is not a straightforward marker. In addition, during the time we were examining the ability of RapaBlock to prevent immune suppression by RapaLink-1, several collaborators working on a completely different animal model (experimental autoimmune encephalomyelitis, EAE) showed that RapaLink1 alone was not immune suppressive in the disease model further complicating the issue of immune suppression. In short, we realized that immune suppression by mTOR inhibitors is a highly complex problem that we could not fully address here. We underestimated the complexity, and we thank the reviewer for his/her comments. We have revised our text to more accurately reflect the limited nature of the PBMC assay and removed the claims regarding immune suppression.

(2.) The rationale to focus on mTOR inhibition in GBM is not sufficiently clear. There is no data to support the claim that “mTOR inhibitors have shown efficacy in GBM” (page 3). In fact, none of the (multiple) clinical trials with rapalogs or mTOR kinase inhibitors have shown clinical activity in GBM and it is far from clear whether this failure is due to inadequate mTOR kinase inhibition (versus feedback-reactivation of AKT, mTOR independent signaling pathways, other). The U87 GBM cells line used in the current study is known to be exceptionally responsive to mTOR inhibitors, but has not been predictive for the experience with mTOR inhibitors in GBM patients. More extensive in-vivo studies using multiple different orthotopic GBM PDX models, which faithfully represent the invasive growth and genetics of human GBM, are required to show that: (1.) mTOR inhibition has antitumor activity which is closely linked to the degree of mTOR inhibition in tumor cells, and that (2) the combination of Rapamycin or RapaLink-1 with RapaBlock is at least some ways “better” (more antitumor activity or less systemic toxicity) than Rapamycin or RapaLink-1 in the absence of RapaBlock. Show that the addition of Rapablock to RapaLink-1 does not reduce the antitumor activity of RapaLink-1 is important, but not compelling enough.

We thank the reviewer for pointing out the limitations of U87 cells and suggesting patient derived xenograft models. We have performed in vivo studies using an orthotopic PDX GBM model (GBM43 cells) which is a very aggressive tumor model. We found that the RapaLink-1+RapaBlock combination conferred a survival benefit compared to RapaLink-1 alone or vehicle. In this aggressive model, other readouts such as bioluminescence and body weight were not as pronounced as we observed with U87 cell model. This data is now included in Figure 4c.

Referee #2 (Remarks to the Author):

This is an exciting study that exploits the surprising finding that effective cellular engagement of kinases by bivalent molecules composed of an ATP – competitive kinase inhibitor linked to FKBP12 binders requires successful engagement of FKBP12. The authors show that this activity can be exploited to achieve brain selective pharmacology by blocking effective engagement in tissues with a FKBP12 blocking agent. Impressively the authors demonstrate proof of concept for four different kinases showing the potential generality for the approach. Overall, I find this to be a very exciting study and the conclusions are well supported by the presented data and I think it is meritorious of publication. A couple of minor points to address:

1. The authors convincingly demonstrate the requirement for FKBP12 engagement to achieve cellular potency through FKBP12 knock-down and chemical competition experiments. Another useful control would be to show a negative control compound containing a subtle chemical change to the FKBP12 binder that reduced FKBP12 binding and examining its effect on potency. I would assume the authors have prepared such a compound for one of their test systems.

This is a great point. We have now prepared an oxime derivative (Juvvadi et al. PMID 31537789) of RapaBlock, which has reduced binding (360 nM) affinity to FKBP12 compared to RapaBlock (1.4 nM). RapaBlock-Oxime was much less effective at blocking the inhibition of mTOR by RapaLink-1. This data is now included in Extended Data Figure 4.

In addition, the SLF series of analogs described in the paper, which have 10-100 fold lower

binding affinity to FKBP12, also serve as such “negative controls”. The SLF analogs are

uniformly worse at blocking rapamycin and RapaLink-1 (see Extended Data Figure 2). For example, SLF, which has a K_i (FKBP12) of 25 nM (for reference, RapaBlock has a K_i of 3.1 nM), can barely block the mTOR inhibition by RapaLink-1 (Entry 1, Extended Data Figure 2a).

2. I didn't see any pharmacokinetic data for the rapablock compound. It would be good to show PK/PD relationship in peripheral tissues to establish that drug exposure in the periphery by rapablock is indeed required to suppress the pharmacology of the Rapa-link mTOR inhibitor.

We have attempted to measure the serum half-lives of RapaLink-1 and RapaBlock in animals. However, the unusual FKBP12-binding property of these two compounds makes serum drug concentration an inaccurate metric for pharmacologically relevant intracellular drug level. For example, a previous study (Marinec 2009, 19164520) has revealed an intracellular sequestration effect for FKBP12-binding drugs, where the serum concentrations of such drugs are much lower than the intracellular concentrations. The high cellular concentration of FKBP12 (~10 μ M) further magnifies the challenge to accurately measure intracellular drug concentration. We have therefore relied on the well validated pharmacodynamic markers of mTOR inhibition. We hope the reviewer agrees that in this case, analysis of mTOR activity by Western blot, as well as the chronic experiments, is a more sensitive and pharmacologically relevant readout of the effective drug concentration.

3. I didn't understand why the authors pursued macrocyclic rapamycin/FK506-kinase inhibitor hybrids rather than the smaller and synthetically more tractable SLF type-kinase inhibitor hybrids. Perhaps this could be explained more clearly in the text.

This is another excellent point related to Question 1. The primary issue with SLF type FKBP ligands is their inferior affinity to FKBP12, which would be detrimental to FKBP12-mediated intracellular partitioning. We did in fact synthesize a hybrid SLF-Dasatinib compound. Unfortunately, this compound did not have kinase inhibitor activity in cells (see data below). We suspect that its low affinity to FKBP12 is a main contributor to the lack of activity. Because of this, this compound was not the main focus of the manuscript, and we did not perform additional studies and minimized speculative discussions.

Drug (100 nM) DMSO Dasa FK-Dasa SLF-Dasa

Jurkat cells stimulated by anti-CD3 (OKT3, 5 μ g/mL) for 5 min and analyzed by immunoblot.

4. The authors should show that rapablock is pharmacologically silent (cell proliferation, RNA sequencing etc) if this agents is really just suppose to be a blocking agents.

This is an important question. We believe that the current proliferation data in the manuscript (Figure 3b, 3e) already shows that RapaBlock alone does not affect cell proliferation (red curves, where a constant 10 μ M RapaBlock is included in the assay). The data points with low RapaLink-1/rapamycin concentrations had ~100% relative proliferation compared to DMSO-treated cells.

In addition, we would like to point to additional data from our lab's recent publication using the RapaLink-1/RapaBlock combination in the setting of alcohol use disorder (Ehinger et al. Nature Communications, 2021), where we showed that RapaBlock treatment *in vivo* was not toxic and had no effect on sophisticated mouse behavioral assays related to alcohol self-administration. We have included this citation in the text.

Referee #3 (Remarks to the Author):

Summary of the key results

Zhang et. al. introduce a novel strategy to achieve brain-specific inhibition of mTOR, while sparing mTOR activity systemically. The authors use a brain-permeable mTOR inhibitor RapaLink-1 and brain impermeable FKBP12 ligand dubbed RapaBlock. The approach is designed to mitigate systemic side effects of mTOR inhibitors when targeting brain cancer or neurological disorders such as TSC. The approach might increase the therapeutic window of FKBP12-binding drugs for the treatment of central nervous system diseases, such as RapaLink-1.

Originality and significance

The approach is novel and has landmark character

Data & methodology: validity of approach, quality of data, quality of presentation

The presentation and interpretation need improvement (see below).

The methodology is sound, the description and (chemical) characterization is lacking in part (see below)

Appropriate use of statistics and treatment of uncertainties

Number of experiments and quantifications need improvement (see below)

Conclusions: robustness, validity, reliability

The major conclusions are robust, some parts might be refined and put into a broader context (see below).

References: appropriate credit to previous work?

Some suggestions below.

Clarity and context: lucidity of abstract/summary, appropriateness of abstract, introduction and conclusions

Minor Revisions needed.

Suggested improvements: experiments, data for possible revision

Major points:

1) Introduction: the sentence "Although mTOR inhibitors have shown efficacy in treating a number of central nervous system (CNS) diseases including tuberous sclerosis complex (TSC)^{3,4}, glioblastoma (GBM)^{5,6} [...]" needs clarification. Are the authors referring only to allosteric inhibitors (rapamycin/rapalogs) or also to ATP-competitive inhibitors? In addition to allosteric mTORC1 inhibitors, ATP-competitive molecules are currently investigated in

preclinical studies for the treatment of Tuberous Sclerosis complex (for example <Rageot, 2018, 30359003>; references indicated as . Regarding glioblastoma, AZD2014 has been recently evaluated in a Phase 1 clinical trials [NCT02619864].

We thank the reviewer for pointing out the unclarity and mentioning the relevant clinical trials. We have revised the text to clarify that we refer to both allosteric and orthosteric inhibitors. We have included the references mentioned.

2) Figure 1 (and text line 61ff): the nature of the in vitro kinase assay should be mentioned in the figure legend. The link to Thermofischer does not qualify as “methods” and is not functional. That FKBP12 does not affect the Z'-LYTE assay (?) is not surprising, as small peptides are used as substrates. It seems that the available literature explains the observed data concerning in vitro and cellular results presented here. The authors should put this in a better perspective.

We have edited the figure legend to indicate the nature of the in vitro kinase assays. We have also included official copies from Thermo Fisher of the full protocol for the Z'-LYTE assay used.

We thank the reviewer for pointing out existing literature concerning in vitro and cellular results. We have cited papers that studied or discussed the “intracellular sink” effect of rapamycin and other FKBP-binding compounds. However, to the best of our knowledge, we are not aware of studies that quantitatively measured the mTOR inhibition IC50 values of RapaLink-1 in a cell-free system, with and without added FKBP12.

We would be more than happy to include any references that we inadvertently missed.

3) Line 77ff. The FRB-Rapa(link)-FKBP12 sandwich is certainly important as a “intracellular sink for RapaLink-1 to accumulate in the cell.” Instead of Refs 19,20 biochemical work should here be considered illustrating the “quasi-irreversibility” of the ternary complex. There is literature explaining the effect of rapamycin and FKBP12 blocking S6K, S6, 4EBP, etc. phosphorylation by TORC1 elucidating mechanistical aspects.

We thank the reviewer for affirming our conclusion on the role of FKBP12 as an “intracellular sink”. We have indeed taken advantage of the high affinity and the kinetic stability of ternary complex.

Again, we would be more than happy to include any references that we inadvertently missed.

4) The above discussion of the “intracellular sink” is also important for the key message of the work, the CNS action of Rapalink-1: Results concerning pharmacokinetics (PK) in the RapaLink-1 and RapaBlock context are not presented here, and are also not established in <Fan et al., 2017, 28292440> cited for Rapalink-1 brain penetration. It should be clearly distinguished between pharmacodynamics (PD) and PK. Ideally the authors would present plasma and brain levels including “fraction unbound” (fu) values for RapaLink-1 and RapaBlock, integrating these values to better illustrate the physiologic mode of action of the RapaLink-1 and RapaBlock. It is likely that the observed PD is dominated by the very high affinity of RapaLink-1 to TORC1, and not its actual “brain permeability” (which is probably comparable to other rapalogs).

This is an important point. We have worked hard to measure the PK of RapaLink-1 and RapaBlock in animals. However, as the reviewer pointed out, the unusual FKBP12-binding property of these two compounds makes plasma and brain drug concentrations inaccurate metrics for intracellular drug levels that are pharmacologically relevant. For example, a previous study (Marinec 2009, 19164520) has revealed an intracellular sequestration effect for FKBP12-binding drugs, where the serum concentrations of such drugs are much lower

than the intracellular concentrations. Our attempts at measuring the PK were challenging when we encountered technical problems with detecting low concentrations of RapaLink-1 (administered at only 1 mg/kg or 1.2 mg/kg).

We completely agree with the reviewer that the very high affinity of RapaLink-1 to TORC1 is an important contributor to the brain PD we observe, and that a comprehensive PK characterization would be helpful in understanding the physiological mode of action. However, we hope the reviewer understands that due to the unusual properties of RapaLink-1 (as the reviewer pointed out), a full PK study will require development of reliable extraction methods to distinguish intracellular and extracellular drugs in various tissues as well as sensitive detection methods to quantify low RapaLink-1 concentrations. We regret that this falls out of our current capabilities.

We have therefore relied on the well validated pharmacodynamic markers of mTOR inhibition. We hope the reviewer agrees that in this case, analysis of mTOR activity by Western blot is a more sensitive and pharmacologically relevant readout of the effective drug concentration.

5) A conclusive demonstration that RapaBlock is not immunosuppressive is not really provided, although the authors results demonstrate sparing of TORC1 signaling. The authors should either provide a more differentiated text (not referring to mTOR PD experiments as “non-immuno-suppression”, or add real *in vivo* data comparing RapaLink plus/minus RapaBlock in a relevant immune response.

1. This is an excellent point – we thank the reviewer for pointing this out. We agree with the reviewer that the *ex vivo* PBMC proliferation data does not fully address the immune suppression issue. We have performed extensive *in vivo* experiments (T cell counting, various T cell, B cell, etc. as well as mass cytometry with over 40 cell surface markers) using Rapamycin as a control, along with RapaLink-1, RapaBlock, and RapaLink-1 + RapaBlock. Frustratingly, we could not obtain convincing positive control data of immune marker suppression or activation of regulatory T cell markers with Rapamycin in a robust manner. Reading through the Rapamycin immune suppression literature it became clear that there is not a straightforward marker. In addition, during the time we were examining the ability of RapaBlock to prevent immune suppression by RapaLink-1, several collaborators working on a completely different animal model (experimental autoimmune encephalomyelitis, EAE) showed that RapaLink1 alone was not immune suppressive in that model, further complicating the issue of immune suppression. In short, we realized that immune suppression by mTOR inhibitors is a highly complex problem that we could not fully address here. We underestimated the complexity, and we thank the reviewer for his/her comments. We have revised our text to more accurately reflect the limited nature of the PBMC assay and removed the claims regarding immune suppression.

6) Extended Data Figure 2: a legend on how to apply the color code should be included to improve clarity. The statement “none of the SLF derivatives were found to be effective”? should be further elucidated. SLF derivatives have a lower affinity to recombinant FKBP12 with respect to FK506 analogs, but some of them showed efficacy in blocking mTOR inhibition by rapamycin.

We thank the reviewer for pointing this out. We have modified the figure to include a color scale that explains the color coding.

The reviewer is correct about the ability of SLF analogs to block rapamycin. As our primary goal is to block RapaLink-1 (which is a more potent inhibitor of mTOR), we chose to focus

our efforts on FK506 analogs that are more effective blockers. We agree that the original statement is not fully clear and have modified the text to differentiate the effects on rapamycin and RapaLink-1.

7) Authors should comment on the correlation between blocking of mTOR inhibition by rapamycin and RapaLink-1? Some compounds showed a different behavior (e.g. 03-087). What trend was observed (compounds more effective in blocking rapamycin action) and what is the mechanistic basis?

Thank you for this suggestion. We have modified the text to differentiate the effects of the compounds on rapamycin and RapaLink-1.

We agree with the reviewer that the general trend is that compounds are more effective in blocking rapamycin than RapaLink-1. Compound 03-087 also follows this trend. We are confident that it is because RapaLink-1 has a higher affinity to mTOR (as part of a ternary complex) due to its bivalent nature (avidity effect). That said, unambiguously establishing the mechanistic details will require extensive biochemical characterization in a three-component system. As the focus of this manuscript is the pharmacological effects of RapaBlock and its ability to achieve brain-specific mTOR inhibition by RapaLink-1, we choose to present our phospho-S6 screen data without adding speculative discussion. We hope the reviewer understands.

8) What strategy has guided the selection of the four RapaLink-1 blocking compounds? The authors should clarify the threshold values considered for the selection. Some modelling of equilibria ligand-protein complexes would also be very helpful to demonstrate the action of the RapaLink-1/RapaBlock.

We did not use a threshold for the selection of candidate compounds. Instead, because there are no reliable methods to predict BBB-permeability, we chose four compounds with distinct chemical features (thioether, amino acid, pyridine, pyridine N-oxide, respectively), confirmed their activity in cell culture, and tested them in vivo. Unfortunately, limited by the practical throughput of in vivo experiments, we were not able to evaluate all compounds that were active in cell culture.

9) In extended data Figure 3, the four molecules are 05-026, 05-037, 06-039 and 06-041. However, 06-039 and 06-041 are not listed in Extended Data Figure 2. The chemical structure and the activity of these two derivatives have to be included in Extended Data Figure 2.

We apologize for the confusion. The compounds were numbered based on the notebook page number on which they were prepared. 06-039 is a different batch of preparation but is chemically identical to 05-084. Likewise, 06-041 is identical to 05-086. We have corrected the compound numbering.

10) Among the four selected RapaLink-1 blocking compounds, it looks like there are no pyridine N-oxide derivatives, that would be 05-060, 05-061, 05-086 and 05-092. Could the authors review the numbering and clarify this point?

We did not select the four candidate molecules with a preference for the pyridine N-oxide structure. Rather, we chose these molecules because of their distinct chemical features and we later found that the pyridine N-oxide compound (06-041) had the best blocking effect in vivo.

11) Please include the compounds' numbers in the text description to guide the readers through the results: "A pyridine N-oxide derivative of FK506 (XX-XXX)".

Thank you – we have made this change.

12) In the supporting information, RapaBlock has to be matched with the code of the main text (05-086).

Thank you – we have made changes. Please also see notes in comments 9 and 11.

13) Figure 4: the labelling is too small and needs to be increased. The gray color of the vehicle is difficult to see.

We have enlarged the figures as permitted by journal graphical guidelines. We have also changed the light grey color using in the vehicle group to a darker color.

14) Same Figure: the quantification for the Western blot showing that “while mTOR activity in the brain was inhibited at a comparable level to mice treated with RapaLink-1 only, mTOR activity in skeletal muscle was not affected” needs to be included. Western blots should be quantified. n=1 and n=2 experiments are problematic for useful (non-parametric) statistics.

We have now quantified the western blot and included the density values under the blots.

15) Figure 5 and associated text. The other FK-drugs are “sunked” into cells, but the overall mode of action is not comparable with the RapaLink approach, where both binding sites are in the same complex (TORC1). In this respect the section is an interesting outlook to “brain-specific” action of target, but the characterization of the FK-compounds is a bit scarce.

We thank the reviewer for recognizing the potential of these FK-drugs, and we apologize for the relatively short discussion due to the limit on manuscript length. We chose to include the characterization data for these compounds in the Extended Figures. We included four different inhibitors to illustrate the generality of our approach, and we acknowledge that the HGK inhibitor has not been evaluated in cellular experiments due to our limited capacity and lack of expertise in neuronal cell cultures.

16) Extended Data Figure 5: Panel g and h need clarification both in the main text and in the Figure legend. What is “1” standing for? Is “Dasatinib and 1” meaning to “Dasatinib and FK-Dasatinib”? The authors state that identical spectrum of kinase targets was observed for Dasatinib and FK-Dasatinib, with the exception of DDR2. Where could this be observed in Extended Figure 5? Labeling for DDR2 is missing.

The figure legend has been corrected in an updated manuscript we submitted to the editor on 11/24. We realized the original text was inaccurate. We have 1) provided original kinase profiling data as a supplemental spreadsheet, 2) modified the language to state that “Among the reported targets of Dasatinib, Src family kinases were potentially inhibited by both Dasatinib and FK-Dasatinib, while a few tyrosine kinases (e.g. DDR1, DDR2) exhibited differential susceptibility to these two inhibitors.” We chose DDR2 for a dose-response study.

In the original Extended Fig 5g, DDR2 was 51% inhibited by Dasatinib and 10% inhibited by FK-Dasatinib. It appeared in the lower middle portion of the figure and was not labeled. We have now added labels for DDR1 and DDR2. We are unable to label all dots in Extended Fig 5g; instead, we have provided the full table of kinase profiling results as Supplementary Data. Because DDR2 was not captured by the occupancy probe XO44, it does not appear in Extended Fig 5h.

Minor points:

17) Fig. 5: choose a color different from gray for the circles (badly visible in the printed version).

The color appeared gray likely due to the thin line width. We have boldened the lines.

Formatting and typos:

18) remove brackets: In "(K562/dCas9-KRAB cells)" \diamond In K562/dCas9-KRAB cells

Thank you. We have corrected this error.

19) Figure 2 legend: "an FKBP12-dependent mTOR inhibitor" \diamond "a FKBP12-dependent mTOR inhibitor"

Thank you. We defer to the opinion of the editors whether "an" or "a" should be used before an acronym with a vowel sound.

20) Increase the resolution of Extended Data Figure 1 (including the resolution of the chemical structure RapaTAMRA).

We apologize for the compression of figures during manuscript submission and conversion. The figures were provided separately as high-resolution PDFs.

21) The synthesis and characterization of RapaTAMRA should be included in supporting information.

We have now included synthesis and characterization of RapaTAMRA in Supplementary Information.

In Supporting information:

22) Add the synthesis and characterization of RapaTAMRA.

Thank you. We have now included synthesis and characterization of RapaTAMRA in Supplementary Information.

23) Are the product UV active after the addition of the pyridine moiety? Can HPLC be used to determine the purity of the final compounds?

These compounds are weakly UV-active and can be detected if we inject a large amount. However, the analysis of FK506 derivatives by HPLC is complicated by the presence of rotamers and tautomers in aqueous solutions, whose ratios are condition-dependent (see: <https://pubmed.ncbi.nlm.nih.gov/8851758/>). For these reasons, we were unable to assess the purity of FK506 derivatives by HPLC. We hope the reviewer understands.

24) S1 and S2 have to be related to the numbering reported in the main text.

Thank you. We have taken additional edits to ensure that the compound numbering remains consistent throughout the manuscript. Synthetic intermediates that do not appear in the main manuscript or extended data figures were numbered with temporary identification numbers in the order of appearance in the Supplementary Information. We have added explanatory text in the Supplementary Information to indicate how compounds are numbered ("Numbering of Compounds").

25) The IUPAC name or SMILE code should be added for the final compounds reported in SI.

Thank you. We have now added IUPAC names for the final compounds and all the synthetic intermediates.

26) The authors state that ¹³C NMR peaks of both rotamers are reported collectively. The number of carbons in S2 is 82 instead of 98 considering the rotamers. Could the authors comment on this? Are there peaks corresponding to different carbons (for example methoxy groups are equal)?

While we have attempted to do so, we could not unambiguously assign each peak in the ¹³C NMR to each rotamer because of the overlapping peaks. Because proton-decoupled ¹³C NMR is not quantitative, we were also unable to discern the peaks by relative intensity. We believe the reviewer is correct that the methoxy groups from both rotamers have identical chemical shifts. However, as the rotamers are not separable, we cannot experimentally confirm this and therefore we chose to report all peaks collectively. We hope the reviewer understands.

27) For the ¹H NMR tabulation of one rotamer, protons are missing e.g. S2 has 74 protons and 69 are tabulated. If not all the protons can be detected (e.g. the OH), it should be stated in the tabulation.

Thank you. In some cases, the presence of both rotamers whose peaks overlap prevents us from accurately determining the proton counts by integration. This is exacerbated by the large number of protons (>70) and the close ratio (3:2) of the two rotamers. We made our best effort to report the peaks and the nuclide counts from the major isomer, but for some compounds it was not practically possible to account for all the expected protons. We have now added these statements to the tabulation and the "General Notes" section.

28) According to Nature guidelines on "Characterization of chemical materials", the authors have to "provide adequate data to support assignment of identity and purity _for each new compound described_. For most organic and organometallic compounds chemical identity should be established through spectroscopic analysis. Please provide standard peak listings for ¹H-NMR and proton-decoupled ¹³C-NMR for all new compounds, with other NMR data (³¹P-NMR, ¹⁹F-NMR, and so on) when appropriate. For new materials, we require high-resolution mass spectral (HRMS) data to support molecular weight identity." In case of "Combinatorial compound libraries, descriptions of the preparation of combinatorial libraries should include _standard characterization data for a diverse panel of library members._"

29) The authors need to report the synthesis and characterization of the representative SLF and FK506 derivatives (reported in Extended Data Figure 2).

30) Moreover, the synthesis and full characterization of ATP-site kinase inhibitors linked to FK506 have to be reported in the supporting information since these are new compounds developed as proof-of-concept of the broader feasibility of the proposed strategy.

Specifically, FK-GNE7915, FK-Dasatinib and FK-Lapatinib should be included. Since they are UV compounds, addition of HPLC chromatograms (and purity assessment) is strongly recommended.

We have now included ¹H NMR, proton-decoupled ¹³C NMR and HRMS spectra for the compounds discussed in the manuscript. We have included synthetic procedures for all SLF and FK506 derivatives and representative characterization data. We have included synthetic procedures for FK-GNE7915, FK-Dasatinib and FK-Lapatinib and their characterization data.

For reasons explained in point 23, we were unfortunately unable to assess the purity of FK506 derivatives by HPLC.

31) Ref 31 lacks journal

Thank you – we have fixed this error.

Reviewer Reports on the First Revision:

Referees' comments:

Referee #1 (Remarks to the Author):

The authors provided additional data and addressed my comments. My only remaining suggestion is to tone down the statement in the introduction that "...mTOR inhibitors have shown efficacy in treating glioblastoma (GBM) ...". Despite encouraging data in selected preclinical GBM models, none of the current mTOR inhibitors have shown convincing antitumor activity in GBM patients thus far.

Referee #3 (Remarks to the Author):

R3

Remaining major points: see R3 for reviewer query

2)

Fig. 1 (and text line 61ff): The nature of the in vitro kinase assay should be mentioned in the figure legend. The link to ThermoFisher does not qualify as "methods" and is not functional. That FKBP12 does not affect the Z'-LYTE assay (?) is not surprising, as small peptides are used as substrates. It seems that the available literature explains the observed data concerning in vitro and cellular results presented here. The authors should put this in a better perspective.

AU_2: We have edited the figure legend to indicate the nature of the in vitro kinase assays. We have also included official copies from ThermoFisher of the full protocol for the Z'-LYTE assay used.

We thank the reviewer for pointing out existing literature concerning in vitro and cellular results. We have cited papers that studied or discussed the "intracellular sink" effect of rapamycin and other FKBP-binding compounds. However, to the best of our knowledge, we are not aware of studies that quantitatively measured the mTOR inhibition IC50 values of RapaLink-1 in a cell-free system, with and without added FKBP12.

We would be more than happy to include any references that we inadvertently missed.

R3_2: The question was not only related to the IC50 of RapaLink, but to what RapaLink does to the mTOR complex. The rapamycin-induced FRB-FKBP12 interaction also blocks access for big substrates in a native TORC1 complex, but it is not clear if this is true for the peptides in the Z'-LYTE assay. The fact that RL-1 does not beat mTOR kinase inhibitors (TORKi) in the presence of FKBP suggests that the assay does not represent a measurement for real life activity. There are a number of papers that have studied mTOR-Rapa-FKBP interactions (<<Banaszynski et al., 2005, 15796538>> <<Bayle et al., 2006, 16426976>> <<Wang et al., 2019, 31466449>>, etc.), which suggest that the assay in Fig. 1b does not reflect the real output of TORC1 activity after Rapa or RL-1 binding. However, the assay is likely rather reliable when the ATP-site inhibitors are considered. The results are thus not surprising (line 66ff), and the discussion of this figure (line 61ff) should be adapted.

3)

Line 77ff: The FRB-Rapa(link)-FKBP12 sandwich is certainly important as a “intracellular sink for RapaLink-1 to accumulate in the cell.” Instead of Refs 19,20 biochemical work should here be considered illustrating the “quasi-irreversibility” of the ternary complex. There is literature explaining the effect of rapamycin and FKBP12 blocking S6K, S6, 4EBP, etc. phosphorylation by TORC1 elucidating mechanistical aspects.

AU_3:

We thank the reviewer for affirming our conclusion on the role of FKBP12 as an “intracellular sink”. We have indeed taken advantage of the high affinity and the kinetic stability of ternary complex.

Again, we would be more than happy to include any references that we inadvertently missed.

R3_3:

Again, it is not about references (for the theme see again <<Banaszynski et al., 2005, 15796538>> <<Bayle et al., 2006, 16426976>> <<Wang et al., 2019, 31466449>>, etc.), but about the mechanism of inhibition. The text in line 76ff is misleading: it is likely that the used in vitro assay does not monitor properly rapamycin-mediated TORC1 inhibition. The Rapa-TAMRA seems to support that further, as loss of FKBP attenuates TAMRA retention. The binding models shown should integrate the scheme depicted in Banaszynski et al., 2005 (with values from modern papers), as it clarifies that the Rapa-FKBP complex binds preferably to the FRB domain, and that the rapa-FRB interaction is of lower affinity.

For more comments, please see below.

4)

The above discussion of the “intracellular sink” is also important for the key message of the work, the CNS action of Rapalink-1: Results concerning pharmacokinetics (PK) in the RapaLink-1 and RapaBlock context are not presented here, and are also not established in <Fan et al., 2017, 28292440> cited for Rapalink-1 brain penetration. It should be clearly distinguished between pharmacodynamics (PD) and PK. Ideally the authors would present plasma and brain levels including “fraction unbound” (fu) values for RapaLink-1 and RapaBlock, integrating these values to better illustrate the physiologic mode of action of the RapaLink-1 and RapaBlock. It is likely that the observed PD is dominated by the very high affinity of RapaLink-1 to TORC1, and not its actual “brain permeability” (which is probably comparable to other rapalogs).

AU_4: This is an important point. We have worked hard to measure the PK of RapaLink-1 and RapaBlock in animals. However, as the reviewer pointed out, the unusual FKBP12-binding property of these two compounds makes plasma and brain drug concentrations inaccurate metrics for intracellular drug levels that are pharmacologically relevant. For example, a previous study (Marinec, 2009, 19164520) has revealed an intracellular sequestration effect for FKBP12-binding drugs, where the serum concentrations of such drugs are much lower than the intracellular concentrations. Our attempts at measuring the PK were challenging when we encountered technical problems with detecting low concentrations of RapaLink-1 (administered at only 1 mg/kg or 1.2 mg/kg).

We completely agree with the reviewer that the very high affinity of RapaLink-1 to TORC1 is an important contributor to the brain PD we observe, and that a comprehensive PK characterization would be helpful in understanding the physiological mode of action. However, we hope the reviewer understands that due to the unusual properties of RapaLink-1 (as the reviewer pointed out), a full PK study will require development of reliable extraction methods to distinguish intracellular and extracellular drugs in various tissues as well as sensitive detection methods to quantify low RapaLink-1 concentrations. We regret that this falls out of our current capabilities.

We have therefore relied on the well validated pharmacodynamic markers of mTOR inhibition. We hope the reviewer agrees that in this case, analysis of mTOR activity by Western blot is a more sensitive and pharmacologically relevant readout of the effective drug concentration.

R3_4:

The high affinity of RL-1 is indeed a problem for PK determination and assessment of target occupancy. The provided target evaluation does not describe the PK, but delivers sufficient data for a PD and a demonstration of TORC1 inhibition.

However, to better illustrate the process of RL-1 binding and the systemic protection of TORC1, the authors should provide some model calculations of the two competing reactions of RapaBlock and RL-1 for FKBP, and an estimation for TORC1 binding. K_d values for Rapa binding to FKBP are available, and approximations for Rapa-FKBP to FRB have been determined. This would provide the readers with an understanding for the necessary affinities of FK-drugs and RapaBlock-molecules for FKBP to provide binary targeting.

8)

What strategy has guided the selection of the four RapaLink-1 blocking compounds? The authors should clarify the threshold values considered for the selection. Some modelling of equilibria ligand-protein complexes would also be very helpful to demonstrate the action of the RapaLink-1/RapaBlock.

AU_8:

We did not use a threshold for the selection of candidate compounds. Instead, because there are no reliable methods to predict BBB-permeability, we chose four compounds with distinct chemical features (thioether, amino acid, pyridine, pyridine N-oxide, respectively), confirmed their activity in cell culture, and tested them in vivo. Unfortunately, limited by the practical throughput of in vivo experiments, we were not able to evaluate all compounds that were active in cell culture.

R3_8:

The question did not mainly refer to BBB penetration, but to the comment made in 4): high affinity for FKBP is crucial for the function of the RapaBlock. Some modelling – even if not with highly precise constants – would be appropriate.

In reference to the authors' response: polar surface area, charge, hydrogen bonding, and elevated cLogP would be classical predictors for BBB penetration; MDCK assays could assess whether the

compounds are P-gp substrates.

14)

Same Figure: the quantification for the Western blot showing that “while mTOR activity in the brain was inhibited at a comparable level to mice treated with Rapalink-1 only, mTOR activity in skeletal muscle was not affected” needs to be included. Western blots should be quantified. n=1 and n=2 experiments are problematic for useful (non-parametric) statistics.

AU_14:

We have now quantified the western blot and included the density values under the blots.

R3_14:

The pS6 is crucial for the evaluation of the PD. The available samples (n=3) should be shown as bar graphs with SD or SEM. Null insulin and insulin only can be omitted, but the \pm RapaBlock action should be convincingly demonstrated. The addition of the ratios is insufficient and not Nature journal standard.

Matthias P. Wymann

Author Rebuttals to First Revision:

Referee comments:

Referee #1 (Remarks to the Author):

The authors provided additional data and addressed my comments. My only remaining suggestion is to tone down the statement in the introduction that "...mTOR inhibitors have shown efficacy in treating glioblastoma (GBM) ...". Despite encouraging data in selected preclinical GBM models, none of the current mTOR inhibitors have shown convincing antitumor activity in GBM patients thus far.

Thank you. We have modified the text to say "...mTOR inhibitors have been investigated in a number of central nervous system disease..."

Referee #3 (Remarks to the Author):

R3

Remaining major points: see R3 for reviewer query

2)

Fig. 1 (and text line 61ff): The nature of the in vitro kinase assay should be mentioned in the figure legend. The link to ThermoFisher does not qualify as "methods" and is not functional. That FKBP12 does not affect the Z'-LYTE assay (?) is not surprising, as small peptides are used as substrates. It seems that the available literature explains the observed data concerning in vitro and cellular results presented here. The authors should put this in a better perspective.

AU_2: We have edited the figure legend to indicate the nature of the in vitro kinase assays. We have also included official copies from ThermoFisher of the full protocol for the Z'-LYTE assay used.

We thank the reviewer for pointing out existing literature concerning in vitro and cellular results. We have cited papers that studied or discussed the "intracellular sink" effect of rapamycin and other FKBP-binding compounds. However, to the best of our knowledge, we are not aware of studies that quantitatively measured the mTOR inhibition IC50 values of RapaLink-1 in a cell-free system, with and without added FKBP12.

We would be more than happy to include any references that we inadvertently missed.

R3_2: The question was not only related to the IC50 of RapaLink, but to what RapaLink does to the mTOR complex. The rapamycin-induced FRB-FKBP12 interaction also blocks access for big substrates in a native TORC1 complex, but it is not clear if this is true for the peptides in the Z'-LYTE assay. The fact that RL-1 does not beat mTOR kinase inhibitors (TORKi) in the presence of FKBP suggests that the assay does not represent a measurement for real life activity. There are a number of papers that have studied mTOR-Rapa-FKBP interactions (<> <> <<Wang et al., 2019, 31466449>>, etc.), which suggest that the assay in Fig. 1b does not reflect the real output of TORC1 activity after Rapa or RL-1 binding. However, the assay is likely rather reliable when the ATP-site inhibitors are considered. The results are thus not surprising (line 66ff), and the discussion of this figure (line 61ff) should be adapted.

AU_3:

Thank you.

We agree that the *in vitro* mTOR kinase assay in Figure 1b does not represent the true activity of TORC1 in cells with protein substrates, but it is appropriate for assessment of ATP-site inhibitors. Since the assay is not appropriate for assessment of non-ATP site containing inhibitors like Rapamycin, we have simplified Figure 1b and removed the Rapamycin +/- FKBP12 conditions.

The new description of Figure 1b reads " We performed *in vitro* kinase assays with purified mTOR protein and found that RapaLink-1 exhibited identical IC50 values as MLN0128 (the TORKi portion of RapaLink-1) whether FKBP12 was present or not (Fig. 1b), indicating that RapaLink-1 can engage the active site of mTOR independent of FKBP12 in a cell-free setting."

3)

Line 77ff: The FRB-Rapa(link)-FKBP12 sandwich is certainly important as a "intracellular sink for RapaLink-1 to accumulate in the cell." Instead of Refs 19,20 biochemical work should here be considered illustrating the "quasi-irreversibility" of the ternary complex. There is literature explaining the effect of rapamycin and FKBP12 blocking S6K, S6, 4EBP, etc. phosphorylation by TORC1 elucidating mechanistical aspects.

AU_3:

We thank the reviewer for affirming our conclusion on the role of FKBP12 as an "intracellular sink". We have indeed taken advantage of the high affinity and the kinetic stability of ternary complex.

Again, we would be more than happy to include any references that we inadvertently missed.

R3_3:

Again, it is not about references (for the theme see again <> <> <<Wang et al., 2019, 31466449>>, etc.), but about the mechanism of inhibition. The text in line 76ff is misleading: it is likely that the used *in vitro* assay does not monitor properly rapamycin-mediated TORC1 inhibition. The Rapa-TAMRA seems to support that further, as loss of FKBP attenuates TAMRA retention. The binding models shown should integrate the scheme depicted in Banaszynski et al., 2005 (with values from modern papers), as it clarifies that the Rapa-FKBP complex binds preferably to the FRB domain, and that the rapa-FRB interaction is of lower affinity.

For more comments, please see below.

We completely agree with the reviewer that RapaLink-1 still participates in the FRB-Rapa-FKBP ternary complex, and the biochemical findings with respect to the tripartite nature of Rapamycin inhibition of mTOR still apply to RapaLink-1. Below we include the model requested by the reviewer which follows Banaszynski et al. 2005.

4)

The above discussion of the "intracellular sink" is also important for the key message of the work, the CNS action of Rapalink-1: Results concerning pharmacokinetics (PK) in the RapaLink-1 and RapaBlock context are not presented here, and are also not established in <Fan et al., 2017, 28292440> cited for Rapalink-1 brain penetration. It should be clearly distinguished between pharmacodynamics (PD) and PK. Ideally the authors would present plasma and brain levels including "fraction unbound" (fu) values for RapaLink-1 and RapaBlock, integrating these values to better illustrate the physiologic mode of action of the RapaLink-1 and RapaBlock. It is likely that the observed PD is dominated by the very high affinity of

RapaLink-1 to TORC1, and not its actual “brain permeability” (which is probably comparable to other rapalogs).

AU_4: This is an important point. We have worked hard to measure the PK of RapaLink-1 and RapaBlock in animals. However, as the reviewer pointed out, the unusual FKBP12-binding property of these two compounds makes plasma and brain drug concentrations inaccurate metrics for intracellular drug levels that are pharmacologically relevant. For example, a previous study (Marinec, 2009, 19164520) has revealed an intracellular sequestration effect for FKBP12-binding drugs, where the serum concentrations of such drugs are much lower than the intracellular concentrations. Our attempts at measuring the PK were challenging when we encountered technical problems with detecting low concentrations of RapaLink-1 (administered at only 1 mg/kg or 1.2 mg/kg).

We completely agree with the reviewer that the very high affinity of RapaLink-1 to TORC1 is an important contributor to the brain PD we observe, and that a comprehensive PK characterization would be helpful in understanding the physiological mode of action. However, we hope the reviewer understands that due to the unusual properties of RapaLink-1 (as the reviewer pointed out), a full PK study will require development of reliable extraction methods to distinguish intracellular and extracellular drugs in various tissues as well as sensitive detection methods to quantify low RapaLink-1 concentrations. We regret that this falls out of our current capabilities.

We have therefore relied on the well validated pharmacodynamic markers of mTOR inhibition. We hope the reviewer agrees that in this case, analysis of mTOR activity by Western blot is a more sensitive and pharmacologically relevant readout of the effective drug concentration.

R3_4:

The high affinity of RL-1 is indeed a problem for PK determination and assessment of target occupancy. The provided target evaluation does not describe the PK, but delivers sufficient data for a PD and a demonstration of TORC1 inhibition.

However, to better illustrate the process of RL-1 binding and the systemic protection of TORC1, the authors should provide some model calculations of the two competing reactions of RapaBlock and RL-1 for FKBP, and an estimation for TORC1 binding. Kd values for Rapa binding to FKBP are available, and approximations for Rapa-FKBP to FRB have been determined. This would provide the readers with an understanding for the necessary affinities of FK-drugs and RapaBlock-molecules for FKBP to provide binary targeting.

Thank you. This is a great suggestion. We have performed this modeling, using constants from Banaszynski et al 2005 as well as our own experiments (see figures below and additional details in the new Supplementary Information file). Intracellular FKBP concentration was estimated at 10 μ M (Siekiera et al, 1701173). Intracellular mTORC1 concentration was estimated at 10 nM (Milo et al, 19854939). We make the assumption that free Rapamycin and free RapaBlock can cross the cell membrane, but the FKBP complexes cannot. We believe is a reasonable approximation of the “intracellular sink” effect. To consider the effects of membrane permeability and drug efflux, we assigned a permeability index of 10%, which apply to free drug molecules.

Model – Rapamycin

Assumption: extracellular drugs constitute an infinite reagent reservoir
intracellular free drug concentration 10% that of extracellular because of permeability

How much RapaBlock is needed to block Rapamycin?

Model – RapaLink-1

Assumption: extracellular drugs constitute an infinite reagent reservoir
intracellular free drug concentration 10% that of extracellular because of permeability

How much RapaBlock is needed to block RapaLink-1?

From these modeling results, we can estimate that $>10 \mu\text{M}$ RapaBlock with a K_d 2.5 nM is required to block 50% of mTOR inhibition by Rapa. Conversely, if we use $10 \mu\text{M}$ of RapaBlock, a $K_d < 2.5 \text{ nM}$ is required.

We note that the model has limitations:

- The binding constants used in the model may be subject to systematic and or non-systematic errors. In the figure for Rapamycin above, we used K_{d2} and K_{d4} values from Banaszynski et al. 2005, and K_{d1} and K_{d3} values measured in our lab in the same assay, as we believe this provides a fair comparison. For RapaLink-1, we used the same K_{d2} value and an estimated K_{d4} of 5 nM based on the ATP-competitive inhibitor portion of RapaLink-1.

- We made an approximation for the membrane permeability and efflux susceptibility of the compounds involved.
- We did not consider drug binding to proteins present in tissue culture medium/plasma (e.g. BSA/HSA).
- We assumed that there is an infinite extracellular pool of 10 nM Rapa. It may not be true as the FKBP sink effect could deplete extracellular Rapa.
- The intracellular sink of immunophilins is complicated by the presence of multiple FKBP. There are 14 FKBP which are known to bind Rapa and FK506 based immunophilins (Kozany et al. 2009, PMID: 19418507. DOI: [10.1002/cbic.200800806](https://doi.org/10.1002/cbic.200800806)) and yet only a subset of these support Rapa mediated mTOR (FRB binding) inhibition.

Even with these caveats, the modeling based on Banaszynski et al. 2005 predicts a requirement of $\approx 20 \mu\text{M}$ Rapa-Block to block 2.5 nM Rapamycin or RapaLink-1 which is within 10-fold of our experimental observations.

We thank Dr. Wymann again for suggesting this modeling. We think it is a great addition to the manuscript that is helpful for the broad audience.

8)

What strategy has guided the selection of the four RapaLink-1 blocking compounds? The authors should clarify the threshold values considered for the selection. Some modelling of equilibria ligand-protein complexes would also be very helpful to demonstrate the action of the RapaLink-1/RapaBlock.

AU_8:

We did not use a threshold for the selection of candidate compounds. Instead, because there are no reliable methods to predict BBB-permeability, we chose four compounds with distinct chemical features (thioether, amino acid, pyridine, pyridine N-oxide, respectively), confirmed their activity in cell culture, and tested them in vivo. Unfortunately, limited by the practical throughput of in vivo experiments, we were not able to evaluate all compounds that were active in cell culture.

R3_8:

The question did not mainly refer to BBB penetration, but to the comment made in 4): high affinity for FKBP is crucial for the function of the RapaBlock. Some modelling – even if not with highly precise constants – would be appropriate.

In reference to the authors' response: polar surface area, charge, hydrogen bonding, and elevated cLogP would be classical predictors for BBB penetration; MDCK assays could assess whether the compounds are P-gp substrates.

Thank you. These are great suggestions. We have performed the modeling (see above). The model predicts that as we increase K_d to 25 nM and 250 nM, RapaBlock becomes less effective at blocking RapaLink-1 ($EC_{50} \approx 200 \mu\text{M}$ and $>1 \text{ mM}$, respectively, see figure on the next page), consistent with our observation with the SLF-based ligands. In addition, we have calculated the polar surface area, number of hydrogen bond donor/acceptors and cLogP values and added them to Extended Data Figure 2. While we note that macrocyclic compounds are known to deviate from the classical standard, we believe these numbers are helpful for the readers.

Blocking of 10 nM RapaLink-1 by RapaBlock with various Kd3 values

14)

Same Figure: the quantification for the Western blot showing that “while mTOR activity in the brain was inhibited at a comparable level to mice treated with RapaLink-1 only, mTOR activity in skeletal muscle was not affected” needs to be included. Western blots should be quantified. n=1 and n=2 experiments are problematic for useful (non-parametric) statistics.

AU_14:

We have now quantified the western blot and included the density values under the blots.

R3_14:

The pS6 is crucial for the evaluation of the PD. The available samples (n=3) should be shown as bar graphs with SD or SEM. Null insulin and insulin only can be omitted, but the \pm RapaBlock action should be convincingly demonstrated. The addition of the ratios is insufficient and not Nature journal standard.

Thank you. We have re-quantified the replicates and plotted the bar graph with error bars representing SD as well as individual data points. These are now included in Figure 4a.

Matthias P. Wymann

Reviewer Reports on the Second Revision:

Referees' comments:

Referee #3 (Remarks to the Author):

I have no further queries and thank the authors for their efforts integrate all requests.